# Exploratory studies into seasonal flow forecasting potential for large lakes

Kevin Sene[1], Wlodek Tych[1], Keith Beven[1]

[1]Lancaster Environment Centre, Lancaster University, Lancaster, LA1 4YQ, United Kingdom

*Correspondence to*: Kevin Sene (k.sene1@lancaster.ac.uk)

**Abstract.** In seasonal flow forecasting applications, one factor which can help predictability is a significant hydrological response time between rainfall and flows. On account of storage influences, large lakes therefore provide a useful test case although, due to the spatial scales involved, there are a number of modelling challenges related to data availability and understanding the individual components in the water balance. Here some possible model structures are investigated using a range of stochastic regression and transfer function techniques with additional insights gained from simple analytical approximations. The methods were evaluated using records for two of the largest lakes in the world - Lake Malawi and Lake Victoria – with forecast skill demonstrated several months ahead using water balance models formulated in terms of net inflows. In both cases slight improvements were obtained for lead times up to 4-5 months from including climate indices in the data assimilation component. The paper concludes with a discussion of the relevance of the results to operational flow forecasting systems for other large lakes.

## 1 Introduction

One of the challenges in seasonal flow forecasting is that the lead times of interest often far exceed the hydrological response time of catchments. This means that traditional approaches to data assimilation are often less effective due to the decay in information content at these longer timescales.

However the potential for deriving operationally useful forecasts improves if there are significant storage influences. Perhaps the greatest success to date has been in snowmelt forecasting for basins with a significant winter snowpack and typically this has been based on statistical techniques or sampling of historic records for input to hydrological models; for example using an Ensemble Streamflow Prediction approach (Day 1985, Wood and Schaake 2008). These techniques have also been applied more widely and other more recent developments include the use of seasonal rainfall forecasts, climate indices and ensemble Kalman filter approaches (e.g. Crochemore et al. 2016, Yossef et al. 2016, Huang et al. 2017). However a common finding is that forecast skill may arise as much from the representation of antecedent conditions as from the meteorological inputs, with the balance depending on factors such as lead times and season, as well as location (e.g. Robertson and Wang 2012, Greull et al. 2016, Mendoza et al. 2017).

Another situation where storage influences are important is for large lakes and potential seasonal forecasting applications include assisting with water supply, irrigation and hydropower operations for individual lakes, and for water resources monitoring at a regional or global scale. However some potential modelling challenges are that lake catchment areas may span several climate zones and that monitoring networks are often sparse. Also, with the exception of lake levels and outflows, the main components in the water balance - lake rainfall, tributary inflows and lake evaporation – are often difficult to measure or estimate. In some cases there may also be significant differences between lake and catchment rainfall due to influences on local climate.

For large lakes, perhaps the longest-established operational systems for seasonal forecasting of lake levels are for the Great Lakes in the USA and Canada. Both empirical techniques and physically-based hydrological models are used driven by long-range climate outlooks for precipitation and air temperatures with, more recently, incorporation of ensemble outputs from regional climate models (Gronewold et al. 2011, Bollinger et al. 2017). On a smaller scale, of course, water balance models for reservoirs and regulated lakes often form part of regional water supply models typically using statistical techniques or ensemble streamflow prediction approaches (Yeh 1985, Pagano et al. 2009). The potential for seasonal forecasting using statistical models has also been explored for the rivers downstream of two of the largest lakes in Africa - Lake Malawi (Jury 2014) and Lake Victoria (Siam and Eltahir 2015) - and, over longer time scales, Mulumpwa et al. (2017) derived probabilistic estimates for future levels of Lake Malawi using a univariate stochastic model.

Here we describe exploratory studies into forecasting potential for large lakes using an alternative approach. Following a brief review of the dynamic characteristics of lake response, both stochastic dynamic regression and transfer function approaches are used to explore the relationships between a range of potential predictors and lake levels and outflows. This general approach has been widely applied to real-time flood forecasting applications (e.g. Lees 2000, Smith et al. 2013) and, in addition to the ease with which options can be explored, a key advantage is that few prior assumptions are required about the nature of those relationships (e.g. Beven 2009, Young, 2013).

The methods are evaluated using case studies for Lake Victoria and Lake Malawi including the potential benefits of data assimilation in reducing the impacts of measurement and modelling uncertainties. As in the studies by Jury (2014) and Siam and Eltahir (2015) the use of climate indices is considered since both the El Niño Southern Oscillation and Indian Ocean Dipole are thought to have a significant influence on rainfall in east and southern Africa (e.g. Saji et al. 1999, Nicholson and Selato 2000, Jury and Gwazantini 2002, and Manatsa et al. 2011). Since the main aim was to provide insights into possible model structures, the analyses were based primarily on historical datasets derived as part of previous water balance studies as this allowed a more detailed investigation of lake response than would be possible using contemporary datasets. By chance the periods covered included some of the most significant flood and drought periods on record allowing model performance to be evaluated under these more extreme conditions. The discussion concludes with some suggestions for how the findings could be translated into operational forecasting models.

## 2 Methodology

### 2.1 General approach

The water balance for a lake can typically be expressed as:

$$\frac{dh}{dt} = N(t) - \frac{Q_o(t)}{A(t)} \tag{1}$$

where h is the lake level, t is time, $Q_o$ is the outflow, and A the surface area. The term N is the net inflow, expressed as a volumetric flux per unit area of lake surface, which is sometimes called the net basin supply or freewater and, for a given time interval, is defined as:

$$N = P - E + \frac{Q_c}{A} = \Delta h + \frac{Q_o}{A} \tag{2}$$

Here P is the rainfall on the lake surface, E the lake evaporation, and $Q_c$ is the inflow from the surrounding catchment area, again expressed in terms of unit lake surface area. All terms are averages for the selected time interval. An error term is often included to account for additional terms which normally cannot easily be quantified such as seepage and groundwater inflows at the lake bed, although for simplicity this has been omitted here.

Based on the idealised equations for fluid flow over a weir, the natural outflow from a lake is often expressed in the form:

$$Q_o = ah^b \tag{3}$$

where a and b are empirically derived constants and it is assumed that h is defined relative to a datum value for which outflows are zero. For a rectangular weir, the theoretical estimate for b is 1.5 and in practice values can be estimated either directly from lake levels and discharge measurements or using more approximate techniques (e.g. Skaugen 2004). Furthermore if, as is often the case, the lake area can be assumed to be constant ($A(t)=A_o$), Eq. (1) then reduces to:

$$\frac{dh}{dt} = N(t) - \frac{a\,h^b(t)}{A_o} \tag{4}$$

Some useful insights can be gained by exploring the response for a constant net inflow $N_o > 0$ and integer values of b and, for the present study, as discussed later, the case b=2 is relevant in which case the solution to Eq. (4) can be expressed as:

$$h = h_o \frac{(1+ce^{-t/\tau})}{(1-ce^{-t/\tau})} \tag{5}$$

where $\tau$ is a time constant defined by $\tau = \frac{1}{2}\sqrt{\frac{A_o}{a\,N_o}}$, $c = \frac{h_1 - h_o}{h_1 + h_o}$, $h_1$ is the initial level and $h_o = 2\,N_o\,\tau$ is the equilibrium level (e.g. Sene 2000). During periods of constant net inflow, lake levels therefore tend towards an equilibrium value over timescales which are a function of the net inflow itself, the area of the lake, and the outflow relationship. In contrast, during periods of heavy rainfall, a more rapid response would be expected with levels rising over much shorter timescales.

Generally the relationship between catchment rainfall and runoff is non-linear and affected by catchment antecedent conditions. However, for ungauged catchments it is often found that, on an annual basis, and sometimes a monthly basis, empirical regression relationships can be derived between flow-related parameters of interest and catchment characteristics

such as the area and slope (e.g. Smakhtin 2001). Perhaps the simplest of all such approaches is to assume a mean runoff coefficient $r = Q_{c/}(A_c P_c)$ in which case the net inflow is given by:

$$N = P + k \, P_c - E \tag{6}$$

where $P_c$ is the catchment rainfall, $A_c$ is the catchment area and $k = r \, (A_c /A_o)$.

Equations (5) and (6) together provide a useful – albeit very crude - framework for considering the lake response. That is, during periods of constant net inflow, it might be assumed that the lake outflow is related to the net inflow by a non-linear relationship with a typical response timescale $\tau$. Furthermore if - as is often the case - the variability in evaporation is much less than that in rainfall, then the variations in net inflow might be considered to be primarily a function of the lake rainfall and catchment rainfall. The extent to which these relationships are valid (or not) is explored later.

For the more general case of time-varying net inflows, Eq. (4) can be solved numerically, with the observed levels at the start of each forecast providing initial conditions. For the current exploratory studies a simple iterative solution proved to be sufficient although more computationally efficient solutions could be envisaged. Regarding the estimates for net inflows, one option would be to seek a process based model based on lake rainfall, tributary inflows and lake evaporation estimates. However, due to the difficulties in estimating these components, and the possible interdependence between them, it is often

more practicable to estimate the net inflows from the lake level and outflow terms in the water balance, as indicated by the right hand side of Eq. (2). This is because levels can usually be measured with little difficulty and outflows are often monitored closely, particularly when a lake is important for hydropower generation and/or water supply, as with the present examples.

This approach also has the advantage of avoiding some of the complexities of understanding the water balance but does

require a forecasting model for net inflows and a statistical approach provides one option; for example using the following autoregressive (AR) formulation:

$$y_t \; = -r_1(t)y_{t-1} - r_2(t)y_{t-2}\ldots -r_n(t)y_{t-n} \; + \; e_t \tag{7}$$

where $y_t$ is the dependent variable, $r_i$ are the model coefficients, which can be time varying if required, n is the maximum lag time considered, and $e_t$ is a stochastic noise term. Alternatively if, as might be expected, external influences such as the lake

rainfall are important, then the following linear regression formulation might be considered:

$$y_t \; = \; s_1(t)u_1 \; + \; s_2(t)u_2 + \; \ldots s_m(t)u_m \; + \; e_t \tag{8}$$

where $s_i$ are the model coefficients and $u_i$ are the external input values, such as rainfall or climate indices, lagged by 1,..,m time steps. In contrast the following transfer function formulation allows both serial dependence and external variables to be considered together with a pure time delay $\delta$ if required:

$$y_t \; = \frac{B(z^{-1})}{A(z^{-1})} u_{t-\delta} \; + \; \frac{D(z^{-1})}{C(z^{-1})} e_t \tag{9}$$

where $z^{-1}$ is the backward shift operator ($z^{-i}y_t = y_{t-i}$). The second term represents the residuals of the transfer function input-output model via polynomials **C** and **D** and, although not used for the net inflow component, an Auto Regressive-Moving Average (ARMA) model of this form (e.g. Box and Jenkins 1970) was used in the data assimilation component

described later. Here a single external input is considered but as discussed later the formulation is easily extended to multiple inputs.

The transfer function and data assimilation aspects of the models were implemented using the recursive estimation techniques available as part of the CAPTAIN Toolbox which was developed by Lancaster Environment Centre for operation within the Matlab ® programming environment. These are described in Young et al. (2007) and Young (2011) but in essence provide a range of routines for estimating model parameters and outputs. The stochastic solution techniques used inherently provide recursive estimates of parameters and uncertainty, including how both of these vary over time, as opposed to the simple estimation of posterior means and variances that many other techniques provide.

## 2.2 Case studies

The two lakes considered were Lake Victoria and Lake Malawi which, respectively, are the first and third largest in Africa and lie within the African Rift Valley, which contains a number of other large lakes with both open (with outflows) and closed basins.

At a regional scale, both are economically important since the outflows are harnessed for hydropower generation and to support large-scale irrigation schemes further downstream on the White Nile and Shire rivers. More locally the livelihoods of millions of people are supported through fisheries, water supply, and agriculture. To provide an indication of scale, the combined catchment and water surface area for Lake Victoria exceeds that of countries such as Uganda and Rwanda, whilst the area of Lake Malawi and its catchment is larger than Malawi itself.

For Lake Victoria the lake outlet is just north of the equator and the southernmost part of the catchment is at about $3^{o}$S whilst Lake Malawi extends from about $14^{o}$S at the outlet to $9^{o}$S in the northernmost part of the catchment. The catchment for Lake Victoria lies mainly in Rwanda, Tanzania, Kenya and Uganda whilst that for Lake Malawi is mainly in Malawi and Tanzania. There are also small contributing areas in the Democratic Republic of Congo (for Lake Victoria) and Mozambique (for Lake Malawi).

At their closest points the catchments lie about 500km apart; however they experience markedly different climates in part due to the annual passage of the Intertropical Convergence Zone (ITCZ). For Lake Victoria, which lies fairly centrally within the zone's range, there are two main rainfall seasons and these are typically between March and May and October and December. In contrast, Lake Malawi lies towards the southernmost end of the range resulting in a single main rainfall season from November to April or May in much of the basin, although with some evidence of a temporary reduction in rainfall intensity part way through the season (Nicholson et al. 2014). The predominant climate classifications (Peel et al. 2007) for the lake catchment areas are tropical savannah for Lake Victoria and temperate (dry winters, hot summers) for Lake Malawi, with regions of arid savannah and arid steppe in the south.

Due to topographic influences there are wide variations in annual rainfall within each basin; also both lakes are large enough for the difference between lake water surface temperatures and the surrounding land to affect the local atmospheric

circulation and hence precipitation and evaporation. For example WMO (1983) notes that for Lake Malawi breezes tend to be offshore in the early morning then onshore in the afternoon, leading to 'a preferential tendency for rainfall on the Lake to occur in the early morning rather than the late afternoon'. UNDP (1986) also notes a wind-funnelling affect in the north-western part of the lake due to local topography which can result in annual rainfall exceeding 3000mm in this area, in

contrast to the plateau areas to the west of the lake where values are typically only 700-1000mm. The local impacts are even more pronounced for Lake Victoria and have been the subject of several investigations, including the use of mesoscale models to study the influences on atmospheric circulation both locally (Sun et al. 2015) and regionally (Thiery et al. 2015). Lake inflows generally follow these seasonal trends, although it is worth noting that some of the lake tributaries are ephemeral with flows generally ceasing towards the end of the dry season, particularly in drier parts of the basins.

For both lakes, regular recording of lake levels began in the 1890s and some catchment raingauge observations date back to the period 1900-1910 for Lake Victoria and the 1920s for Lake Malawi. For Lake Victoria monitoring of outflows began in about 1940 whilst for Lake Malawi the first observations began in 1948. Lake outflows have also been regulated for hydropower production from 1953 in the case of Lake Victoria and 1965 for Lake Malawi. However the scheme designs are very different due to the nature of the topography and river channels at the outlet of each lake and some key features include:

• Lake Victoria – the lake outlet used to be at a spectacular natural waterfall until Owen Falls dam was built about 3km further downstream, drowning out the falls; hydropower generation and the lake outflows are now controlled at the dam

• Lake Malawi – outflows are controlled at Kamuzu Barrage more than 80km downstream from the lake outlet, which is possible since the change in elevation is only a few metres between the lake and the barrage. The main hydropower

plants are in natural gorges downstream of the barrage

However, an important point is that the operating rules for both schemes were to a large extent designed to mimic the response of the natural lake, and these are often represented in the form shown in Eq. (3), with values of b close to 2 (e.g. Drayton 1984, Piper et al. 1986). Due to operational requirements, though, there are sometimes minor departures from these rules so a separate outflow record – termed the 'natural flows' here – was derived in which flows were only retained when

similar to those expected from the level-outflow relationships described by Eq. (3). For the Lake Victoria studies, the periods omitted only amounted to a small part of the record (which pre-dates more recent departures) but this occurred slightly more frequently in the case of Lake Malawi, primarily in the later years of the records. During these times there is also an effect on levels although this is much less significant due to the non-linear nature of the outflow relationships.

Table 1 summarises some key characteristics of the long-term water balance for each lake based on previously published

estimates. However, whilst these values are typical, it is worth noting that they can vary significantly between studies depending on the datasets and periods selected and estimation techniques used. Regarding surface areas, estimates also vary although generally the changes with levels are small, for Lake Victoria amounting to about 2% over the historical range of observed levels (e.g. Piper et al. 1986) and for Lake Malawi by less than 1% per metre rise or fall (Lyons et al. 2011). Areas

were therefore assumed constant for these exploratory analyses, although these variations might be included in a more detailed approach.

**Table 1.** Some key physical characteristics of Lake Victoria and Lake Malawi from various sources including Piper et al. (1986), Sutcliffe and Parks (1999) and Sene et al. (2016). All values indicative only

| Key parameters | Lake Victoria (1956-78) | Lake Malawi (1954-80) |
|---|---|---|
| Surface area (km$^2$) | 67,000 | 28,750 |
| Catchment area (km$^2$) | 194,000 | 95,750 |
| Lake rainfall (mm) | 1878 | 1414 |
| Catchment rainfall (mm/year) | - | 1178 |
| Catchment runoff (mm/year) | 343 | 1000 |
| Lake evaporation (mm/year) | 1595 | 2264 |
| Lake outflow (mm/year) | 524 | 418 |

As noted earlier, historical datasets were used and here it is worth noting two landmark hydrometeorological studies in the 1970s and early 1980s (e.g. WMO 1982, 1983). The derived values formed the basis for a number of later studies in which new records and information were added using a wide variety of approaches; see for example Piper et al. (1986), Sene et al. (1994), Sutcliffe and Parks (1999), Nicholson et al. (2000) and Kizza et al. (2012, 2013) for studies on Lake Victoria, plus

the citations therein, and Drayton (1984), Neuland (1984), Jury and Gwazantini (2002) and Sene et al. (2016) for Lake Malawi. The original papers should be referred to for a discussion of the methods and datasets used but, in many cases, the overall approach was similar: namely to reconstruct the rainfall, inflow and evaporation terms from gauges situated around each lake, and in some cases on islands within the lake. In some studies, rainfall-runoff or statistical models were also used to infill or extend tributary inflow records and - in nearly all cases - the analyses were performed on a monthly basis.

A key factor in choosing which records to use here was how well the resulting models described the overall water balance since this provides some confidence in the suitability of the individual components although, as discussed later, over such huge areas estimates can only ever be approximate. Several of the studies cited met this criterion and, primarily on the basis of data availability and completeness, the following estimates were selected:

   • Lake Victoria – monthly estimates from 1925-1978 reported by Piper et al. (1986) and subsequently updated by
Institute of Hydrology (1994) to the period 1925-1990 for lake rainfall (6-8 shoreline raingauges), catchment rainfall (25-30 raingauges) and tributary inflows (up to 20 river gauges) and for 1925-1992 for levels and outflows

   • Lake Malawi – monthly estimates for the period November 1954 to October 1980 reported by WMO (1983) for lake rainfall (16 shoreline raingauges; 1 island gauge), catchment rainfall (53 raingauges) and tributary inflows (about 21 river gauges), and for which aspects of the water balance appear in a number of the studies cited here, such as
Drayton (1984) and Neuland (1984)

The numbers in brackets indicate the approximate numbers of gauges used to derive each component in the water balance in these studies, with ranges provided in some cases due to differing numbers of gauges in different periods. In the Lake Victoria studies, rainfall-runoff models were also used to estimate inflows for ungauged catchments and to help infill missing values whilst, for Lake Malawi, scaling and correlation approaches were used. For the present study, longer term annual level records were also compiled for both lakes from these various sources dating back to the 1890s and overall catchment rainfall estimates derived from individual catchment values using a simple area-weighting approach. Regarding climate indices the following values were used: the Southern Oscillation Index (SOI) (Trenberth 1984), Niño3.4 (NINO34; Trenberth 1997) and the Dipole Mode Index (DMI; JAMSTEC).

For the Lake Malawi records, it is worth noting that sometimes a small correction term is included to account for inflows and losses between the lake outlet and Kamuzu Barrage; however the impacts are small when considered on a monthly basis and as here this term is often omitted. As indicated above, values are for a hydrological year of November to October but, to help comparisons with the Lake Victoria analyses, when describing results the predominant calendar year is cited; for example '1970' refers to the hydrological year 1969/70. For the Lake Victoria datasets that were used another point to note is that, to help to account for the increased rainfall over the lake surface, the lake rainfall estimates based on raingauge observations were increased using linear scaling factors; this was to preserve an overall water balance between the start and end points of the simulation period, although with no constraints on the variability in the intervening years.

As the many citations above indicate, much has been written about the accuracy of the various gauge records available and the derived water balance components, together with issues such as how rainfall measured at the shoreline relates to average rainfall at the lake surface. For these exploratory studies, to facilitate comparisons, unless otherwise stated values for levels and other parameters were generally expressed in standardised form, based on the departure from the mean divided by the standard deviation in each time period of interest. This also helped to reduce the uncertainties in the results by focussing on the underlying signals rather than being concerned about absolute amounts, such as the estimates for lake rainfall. In forecasting applications, another consideration is that real-time observations can be used to adjust forecasts to help to account for modelling and observation errors and the potential value of this process, called data assimilation or adaptive modelling, is discussed later.

## 3 Results

### 3.1 Initial exploratory studies

Figure 1 shows some notable events in the recorded histories for both lakes. These include high levels in the late 1970s and late 1990s and - for Lake Victoria - the most extreme levels on record in the early 1960s and a prolonged period of low levels from the 1920s to the 1950s.

In contrast, for Lake Malawi, levels were unusually low up to the 1930s and several studies (e.g. Drayton et al. 1984) have suggested that this was due to a sand barrier forming at the lake outlet or in the channel(s) downstream in around 1908-1915,

following which levels then rose progressively until the blockage(s) cleared during that decade. However, this event falls outside the period considered here and - with the exception of the minor impacts from hydropower operations noted earlier – for both lakes variations in levels were therefore due primarily to climate influences. In particular the 1961/62 event for Lake Victoria has previously been investigated in detail with some evidence of a regional shift in climate at that time (e.g.

5    Sutcliffe and Parks 1999, Nicholson and Selato 2000).

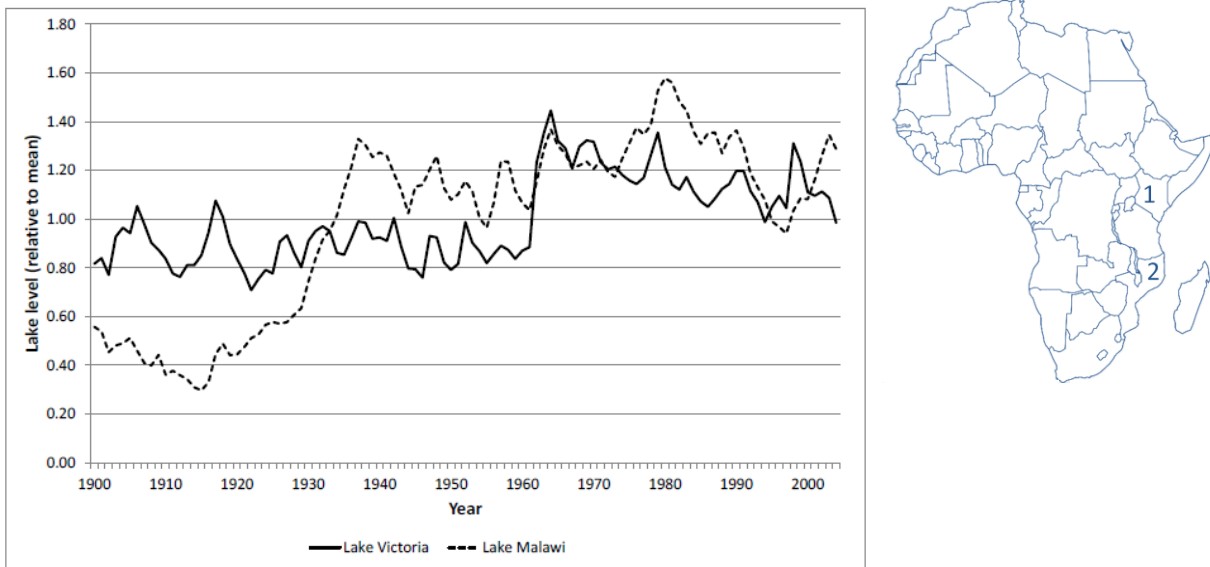

**Figure 1.** Annual lake level variations in Lake Victoria and Lake Malawi from 1900 to 2004 relative to the mean values in that period; values are expressed in terms of indicative depths at the lake outlet. The inset shows the locations of (1) Lake Victoria and (2) Lake Malawi

Another notable feature of the observed levels is the apparent persistence during times of falling levels and the analytical form of the water balance (Eq. (5)) provides some insights into the response during these periods with, as indicated earlier, the use of a constant area and a value of b=2 being reasonable approximations.

    Figure 2 illustrates this response for Lake Malawi for the case of a sudden change in levels, such as might occur following

15    a few weeks of heavy rainfall, or recovery following a prolonged dry spell, and similar response curves have previously been published for Lake Victoria (e.g. Institute of Hydrology 1994) and - using numerical simulations - for Lake Malawi (WMO 1983, Neuland 1984). In both cases, based on typical long-term mean values for the net inflows, the estimated time constants from Eq. (5) were in the range 4-5 years, which may just be coincidence or is perhaps reflective of the balance between net inflows, areas and outflow characteristics required for a lake in this region to have a permanent outflow: a speculative point

20    which might be worth further investigation since there are several other large lakes in the African Rift Valley.

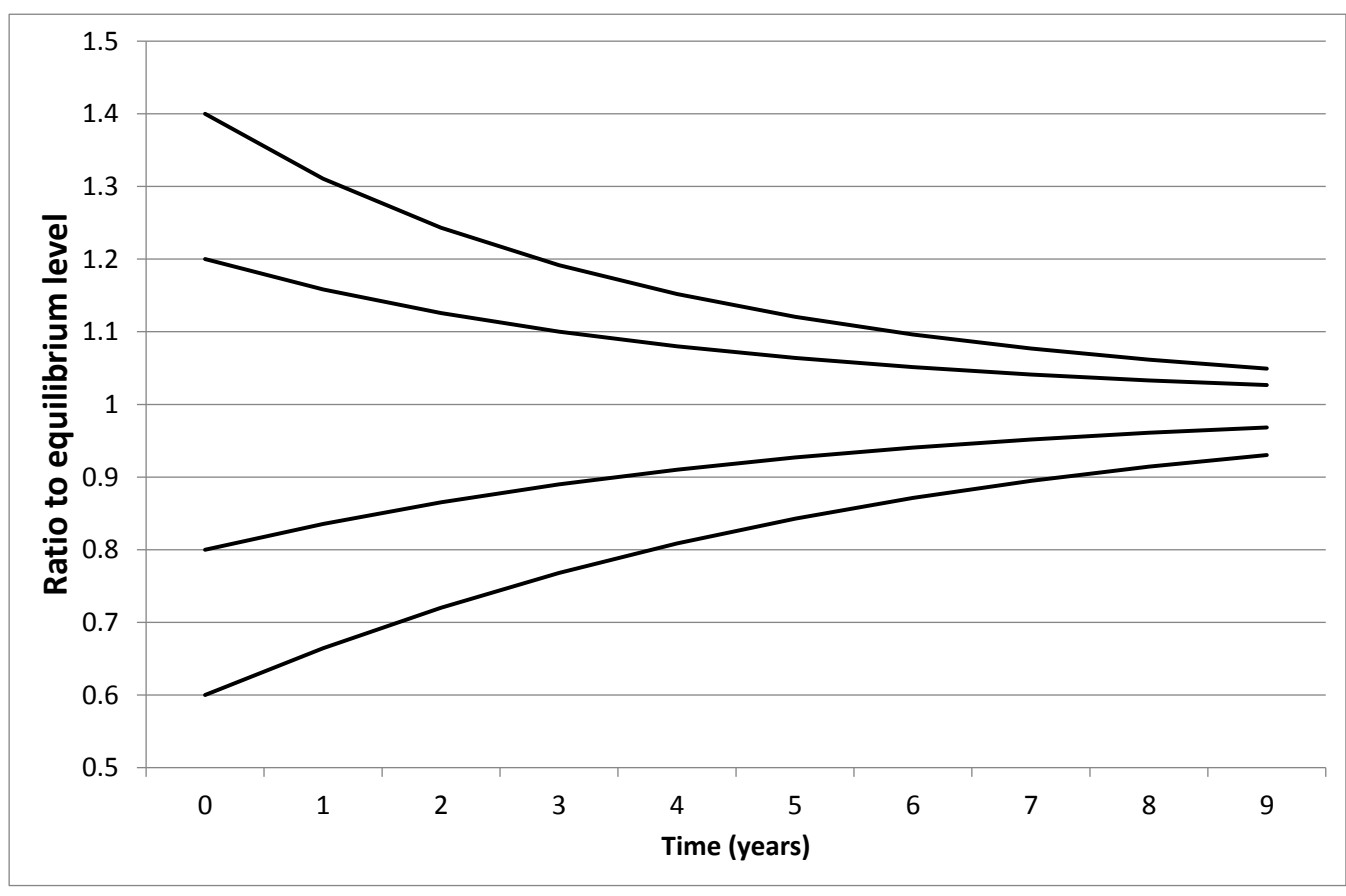

**Figure 2.** Examples of the response in levels for Lake Malawi from initial values equal to 0.6, 0.8, 1.2 and 1.4 times the long-term equilibrium values

More generally these results suggest that, in addition to monthly variations, there are longer-term aspects to the lake response related to both climate variations and the inherent time delays in response. From a forecasting perspective these can potentially be exploited and, to explore these relationships further, both time series and correlation plots were prepared on a monthly basis; in the latter case for a range of assumed lag times. Figure 3 shows one such example, for the case of the Lake Victoria datasets with zero assumed time delay between inputs and outputs. For net inflows, for both lakes the strongest

relationships were with the lake rainfall and catchment rainfall. Here the full records were considered and cross correlation coefficients were in the range 0.82-0.95 at zero time delay and about 0.5-0.8 at a lag time of 1 month for the Lake Victoria and Lake Malawi records, respectively, whilst for tributary inflows the relationships were generally weaker than this. The serial dependence in values was also investigated for some records and typically the autocorrelation coefficients in net inflows were highest for a lag time of 1 month (0.4-0.7 for the two lakes), roughly halving for a lag time of 2 months and

continuing to reduce at longer lag times.

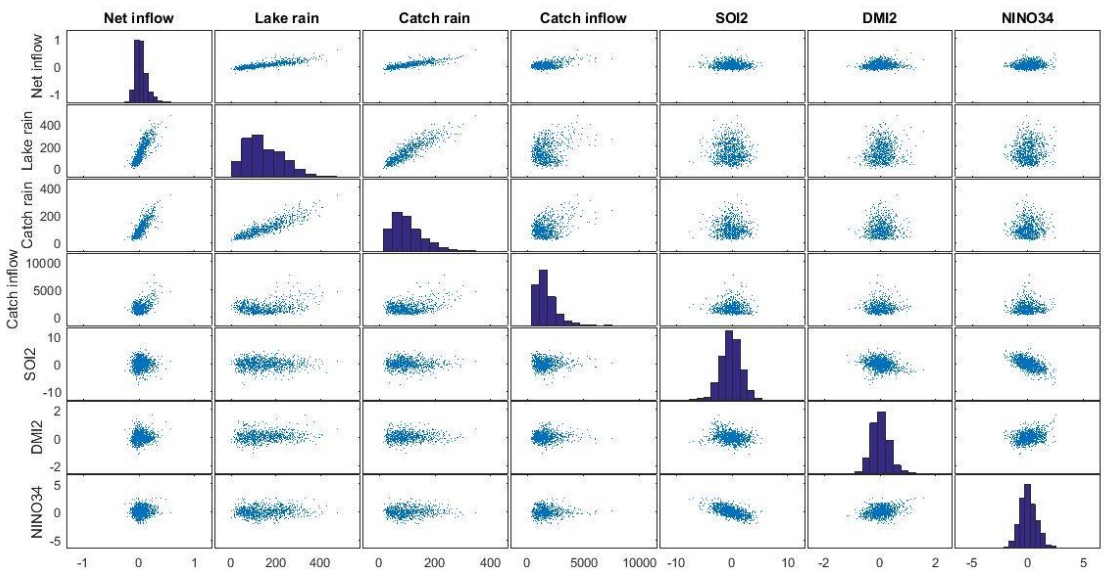

**Figure 3.** Example of a correlation plot for the standardised monthly records for Lake Victoria (1925-1990) assuming zero lag time between values (Catch=catchment)

Regarding climate indices, for both lake records the relationships between net inflow and the El Niño related indices were only borderline statistically significant, with the largest values for lag times of a few months. In contrast the maximum coefficients for DMI were about 0.2 for lag times of 2-4 months. Further investigation showed that for the Lake Victoria records, but not for Lake Malawi, this linkage was markedly higher for the second half of each year – and hence the second rainfall season - reaching values of about 0.27 at lag times of 2-3 months. Interestingly, when compared on a time series

basis, in some periods the correspondence was unexpectedly close; for example during the 1961-1964 event in Lake Victoria the initial peak in net inflows in 1961 was preceded by a rise in DMI a few months before, with similar but smaller rises in advance of peak values for the next three years. However this is just a tentative conclusion and would require further investigation if that event is of particular interest.

### 3.2 Net inflow forecasting

Taken together these initial studies suggested that the following characteristics would provide a useful starting point for developing a model of the net inflows for both lakes:

- Serial correlation – a dependence on values for the past 1-2 months but probably not much beyond that

- Primary external factors – a dependence on current values for lake rainfall and catchment rainfall and possibly for the previous month
- Secondary external factors – inclusion of the Dipole Mode Index (DMI) and possibly one or both of the indices related to the El Niño-Southern Oscillation, at lag times of up to several months

The calibration periods used for model development were 1925-1954 for Lake Victoria and 1954-1970 for Lake Malawi, and these were chosen so that the high levels of the 1960s for Lake Victoria and for the 1970s for both lakes would fall within the validation periods, which were 1955-1990 and 1971-1980 respectively. As noted earlier, standardised values were used throughout at a monthly time step.

Considering the autocorrelation aspects first (Eq. (7)), for the calibration period for the Lake Victoria net inflow record, using a simple autoregressive model the highest values achieved for $R^2$ were about 0.3 with a second order model. The corresponding value for the Lake Malawi record was rather better at about 0.6, again for a second order model.

Regarding external inputs (Eq. (8)), several permutations were considered focussing on the use of rainfall inputs and climate indices. In terms of the $R^2$ performance, the differences between these various regression models were generally not
large, with values typically in the range 0.81-0.94 for both lakes when using time varying parameters. However, based on the simple representation for net inflows shown in Eq. (6), the value of k was similar for both lakes suggesting that a weighted average of these two inputs might also be worth exploring, but the performance was only marginally better than when using lake rainfall alone. The influence of climate indices was also small (about 0.01-0.02 in terms of $R^2$) due to the dominance of the rainfall terms so these were included in the data assimilation components of the models as described later.

On this basis, the decision was taken to use the lake rainfall as the only external input since – as discussed later – this would have some advantages in an operational setting. For the Lake Malawi records, best results were obtained using the latest observed lake rainfall alone whilst - in the case of Lake Victoria - including the previous month's rainfall as well seemed worthwhile. Hence for the Lake Victoria records the expectation was that a [2 2 0] model structure might be appropriate, with a [1 1 0] structure for Lake Malawi, where the notation [n m δ] refers to the parameters in Eq. (7) to (9).

A search of all permutations in the range [1 1 0] to [3 3 3] showed these to be in the top few options in terms of $R^2$ plus a range of other indicators, such as the information criterion described by Young (2011). Recursive estimation of parameter values also showed that these were reasonably stable over time and fixed parameter versions gave similar values of $R^2$; that is about 0.8 and 0.92 for the Lake Victoria and Malawi records respectively. Fixed parameters were therefore assumed for the forecasting runs described later and Fig. 4 shows one such example, for Lake Victoria for part of the validation period.

For the Lake Malawi record the validation performance was similar to that during the calibration period but slightly reduced for the Lake Victoria record. This is perhaps due to the suspected shifts in climate after the 1961 event and, from graphical comparisons, the decrease seemed to be mainly due to slight differences in timing for some years, rather than in magnitudes, as discussed further in the next section.

The estimated confidence intervals are also shown and generally encompassed both the high and low inflow observations, providing some reassurance that the main features of the response are being captured despite the simplifications of fixed parameter values and using lake rainfall values alone rather than a more complex approach. However, at this stage no noise term was included since it proved to be more convenient to include this as part of the data assimilation component.

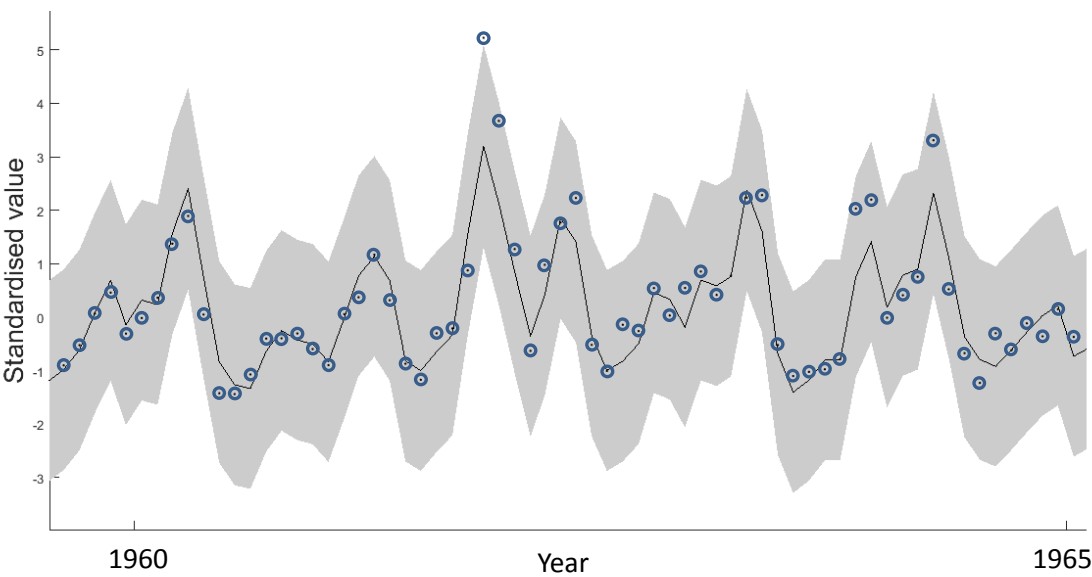

**Figure 4.** Example of standardised net inflow estimates from a transfer function model for Lake Victoria net inflows for part of the validation period, with fixed parameter values; observations appear as dots and the shading corresponds to twice the model standard error, which is approximately equivalent to the 95% confidence interval. Note that, due to sampling uncertainties, the number of values within or outside the intervals varies depends on the period(s) chosen

### 3.3 Forecast performance

Having developed models for the net inflows, these were expressed in recursive form for input to the water balance equation (Eq. (4)). This formulation mimics how the models would be used in an operational setting in which the water balance would be solved numerically each month to derive forecasts for lake levels and outflows for the months ahead based on rainfall observations available up to the time of the forecast. To further extend forecast lead times, rainfall forecasts would ideally also be required but, for these exploratory studies, it was sufficient to use climatological estimates instead, based on the mean monthly distributions of rainfall in the calibration period. However, to provide an indication of forecast potential, the performance was also estimated assuming perfect foresight of rainfall: that is, using historical observed values beyond the

forecast origin. In both cases, the first two years of record were ignored to allow for initialisation of the autocorrelation aspects of the models.

Figure 5 shows the estimated variations in $R^2$ with lead time for the validation periods for both lakes, derived by advancing the forecast origin by one month between each model run and then retrospectively estimating overall values at the required
lead times. The differences between values for levels and outflows are primarily due to the gaps in the derived natural outflow record discussed earlier. As expected, performance decreases with increasing lead times; for example, for lake level estimates falling to about 0.8 after 7 months and 0.9 after about 3-4 months. However, for the Lake Victoria record, due to the rapid increase in levels in the early 1960s, this probably overstates the performance so the values of about 0.7-0.8 at 3-4 months and 0.4-0.6 at 7 months obtained for later years and the calibration period are more typical. This is simply an artefact
of this type of performance measure since using mean values as a reference becomes less informative if a time series is non-stationary, such as exhibiting trends over time and/or quasi-step changes in values as here.

The figures also show the performance for lake levels assuming perfect foresight of rainfall, from which a speculative conclusion might be that, since many seasonal rainfall forecast products tend towards a climatological estimate at long lead times, the values for the control run or ensemble mean might therefore asymptote to those estimates at longer lead times. If
correct, then the differences between the climatological and perfect foresight values provide a rough indication of the potential performance gain from using seasonal rainfall forecasts, with the remaining improvements to be achieved from reducing uncertainties in the models and underlying datasets.

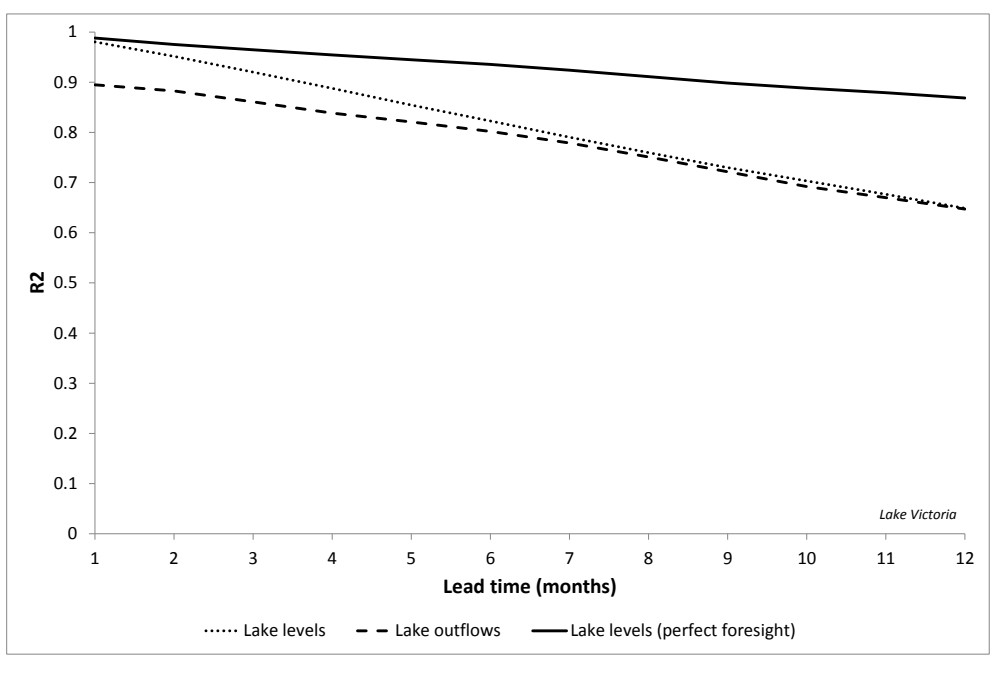

(a)

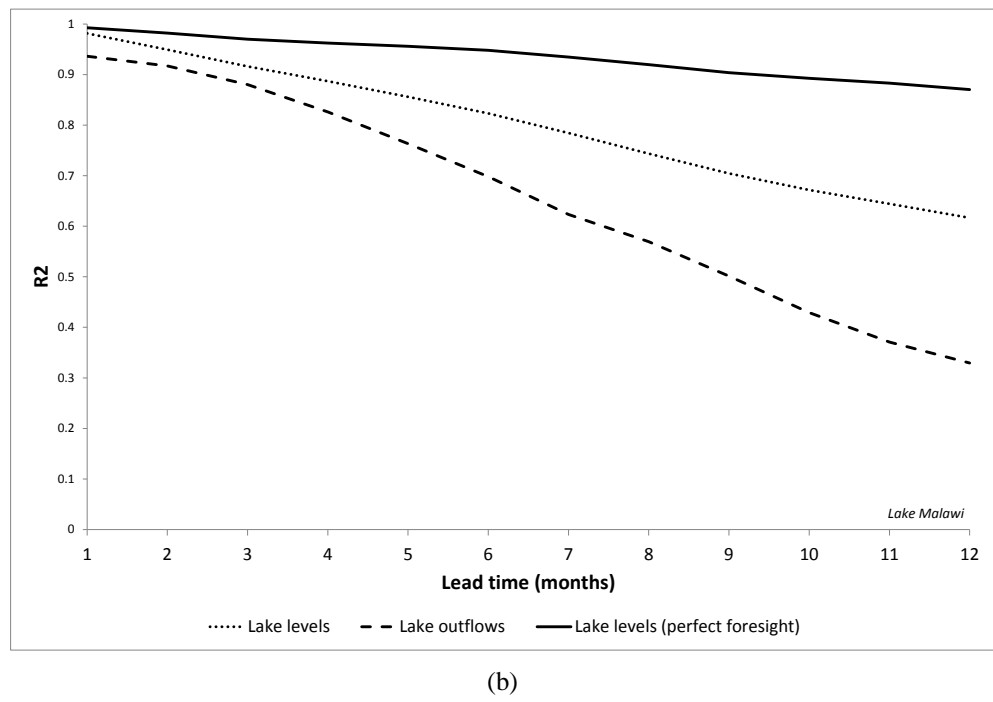

(b)

**Figure 5.** Variations of $R^2$ with lead time for levels and outflows for (a) Lake Victoria and (b) Lake Malawi using climatological rainfall inputs for the validation periods and – for levels – perfect foresight of rainfall

These results of course only give one view of performance and, given the findings from previous climate studies for these lakes, and the cross correlation estimates from the present study, one possibility is that the errors in model outputs might to some extent be explained by longer term variations in climate. From the point of view of developing data assimilation routines, this is a more attractive option than simply developing a statistical model for the residuals since there is then some
10   underlying physical interpretation.

To test this hypothesis, regression models were developed between the residuals of the forecast outputs and the climate indices described earlier, using DMI and NINO34 as examples; similar conclusions were reached using SOI. Both single and multiple regression models were evaluated in time-varying and fixed parameter forms considering a range of possible lag times and forecast lead times. For example, for the Lake Malawi records, for the 4-month ahead forecasts, the maximum
15   cross correlation coefficients with DMI were obtained for lag times of about 3-5 months, and at slightly longer lag times of 6-9 months for NINO34 and SOI, although the signs differed depending on the index chosen. The magnitudes of the coefficients were about 0.42 and 0.28 for DMI and NINO34 in the calibration period, and slightly lower for SOI. For the Lake Victoria record, it seemed useful to consider slightly longer lead times and, for the 6 month ahead forecasts, optimum lag times were about 6-7 months for DMI and 5-6 months for NINO34, with correlation coefficients of about 0.28 and 0.41

respectively, and again slightly less for SOI. As might be expected, these relationships were generally weaker when considering perfect foresight since these influences may already be embedded in the observed data to some extent.

Based on these results, and exploratory studies using single regression models, multiple regression models for lag times of 7 and 5 months for DMI and NINO34 were assumed for the Lake Victoria record and values of 4 and 7 months for the Lake Malawi record. In both cases, the models with time-varying parameters exhibited slightly better performance than their fixed parameter counterparts, with $R^2$ values of about 0.39 and 0.19 respectively for the Lake Victoria regression model and 0.44 and 0.27 for the Lake Malawi model, when using both indices together.

However, the use of $R^2$ only provides one view of performance and, as already noted, has some limitations. Also, lake levels are strongly seasonal and there might therefore be a case to use alternative benchmarks such as seasonally-based values or metrics which incorporate lag times or moving averages as proposed by Schaefli and Gupta (2007). Indeed there are many possible measures which could be used (e.g. Jolliffe and Stephenson 2011, Wilks 2011) and this is a topic that would merit further research. For these exploratory studies, though, a simpler approach was adopted which was to focus on the errors in annual peak levels since these tend to occur around the same time each year and are more challenging to forecast than annual minima.

For the purpose of this exercise alone, separate relationships were also calibrated for the validation period to further explore the strength of these relationships. Fig. 6 shows some example results for the fixed parameter case for the 5-month ahead forecasts of peak annual levels for the Lake Victoria record, and 4-month ahead values for Lake Malawi. The results suggest that in many – although not all – years the adjusted forecasts for maximum levels are closer to those which were subsequently observed, potentially providing a useful gain in forecast performance.

The main exception however was for the Lake Victoria record in the validation period (1955-1990) for which – although negative forecast errors were consistently improved - positive errors were not. Interestingly, the relationship with DMI in that period was also slightly stronger but that with NINO34 no longer significant (<0.1), which is possibly again further evidence of a shift in rainfall response. However, for both records, further investigation would be required into the optimum approach to use; for example, exploring the influence of timing errors on model performance and whether time varying parameters could be used to help represent longer term trends and other variations.

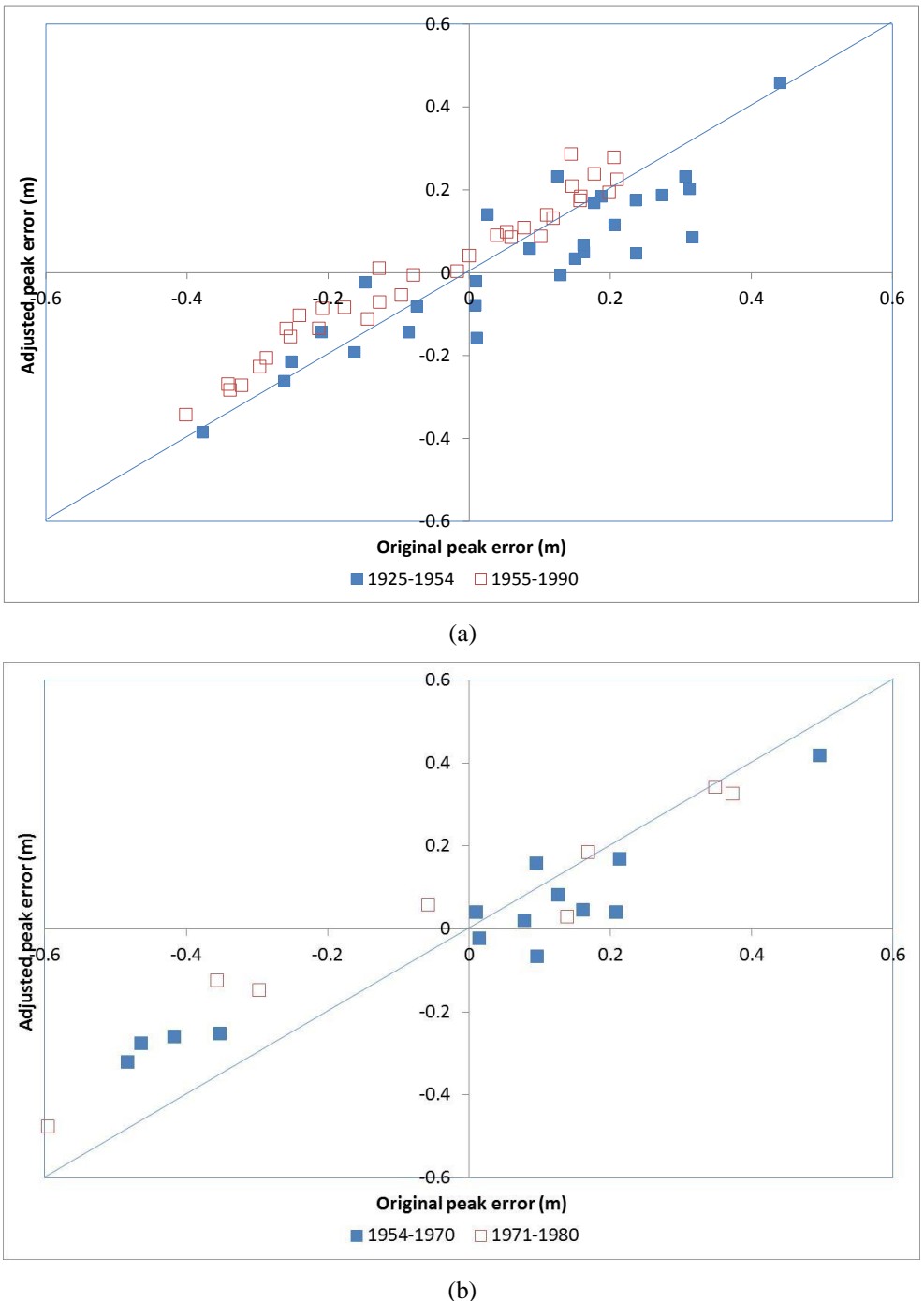

(a)

(b)

 **Figure 6.** Illustration of the effect of including fixed parameter regression relationships with DMI and NINO34 for the model residuals on errors in annual maximum levels at lead times of (a) 5 months for Lake Victoria and (b) 4 months for Lake Malawi. Climatological rainfall inputs were used and a one-to-one trend line is shown as a guide

Given these uncertainties, rather than applying these adjustments first to re-estimate the residuals, the original series were analysed further. Due to lake storage and climate influences, as might be expected the values showed some serial correlation over periods of months superimposed upon longer term variations. However, exploratory studies using an ARMA approach for the residuals – with forecasts implemented via a Kalman Filter - provided mixed results. For example, for the Lake Malawi records, using the same lead time as in Fig. 6 (4-months) and climatological inputs, the forecasts for peak annual levels were only improved in a few years, and even degraded in some cases. As is common with error prediction routines, the issue here seemed to be partly due to timing differences in the residuals, although this problem seemed to be reduced to some extent when assuming perfect foresight of rainfall, again suggesting that reducing timing errors at the outset may assist with performance.

Various permutations of model orders and lead times were explored and Figure 7 shows some examples using the same structures for each lake record, namely (AR(6), MA(5)) when using climatological inputs and (AR(3), MA(2)) for perfect foresight inputs. The lead times used were 2 months for the Lake Victoria record and 4 and 2 months respectively for the Lake Malawi record. Again, for illustration, the results for the full record lengths are shown although the ARMA models were calibrated just for the calibration periods.

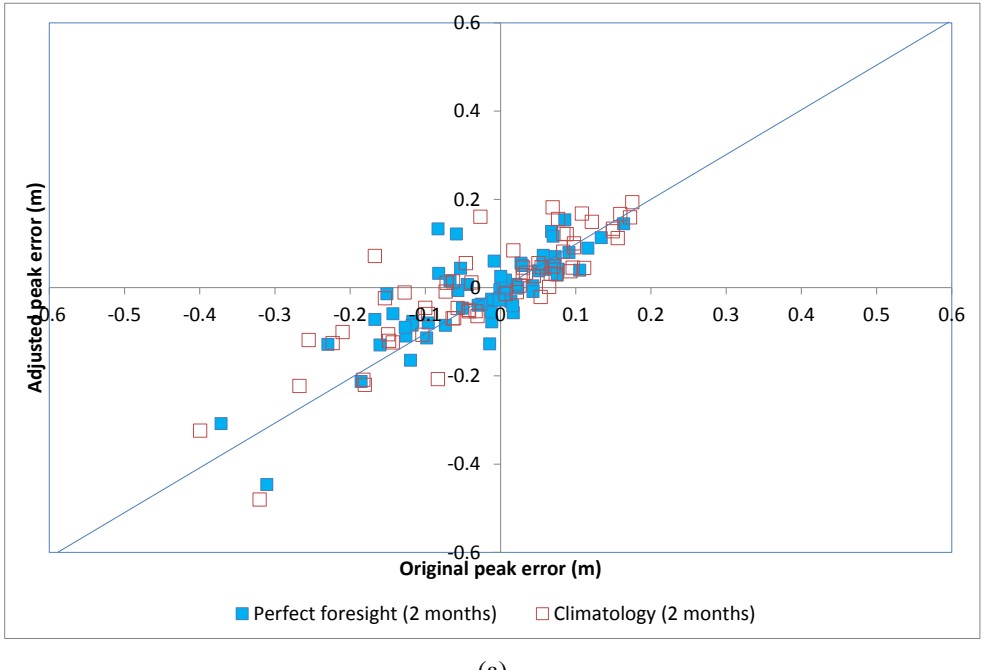

(a)

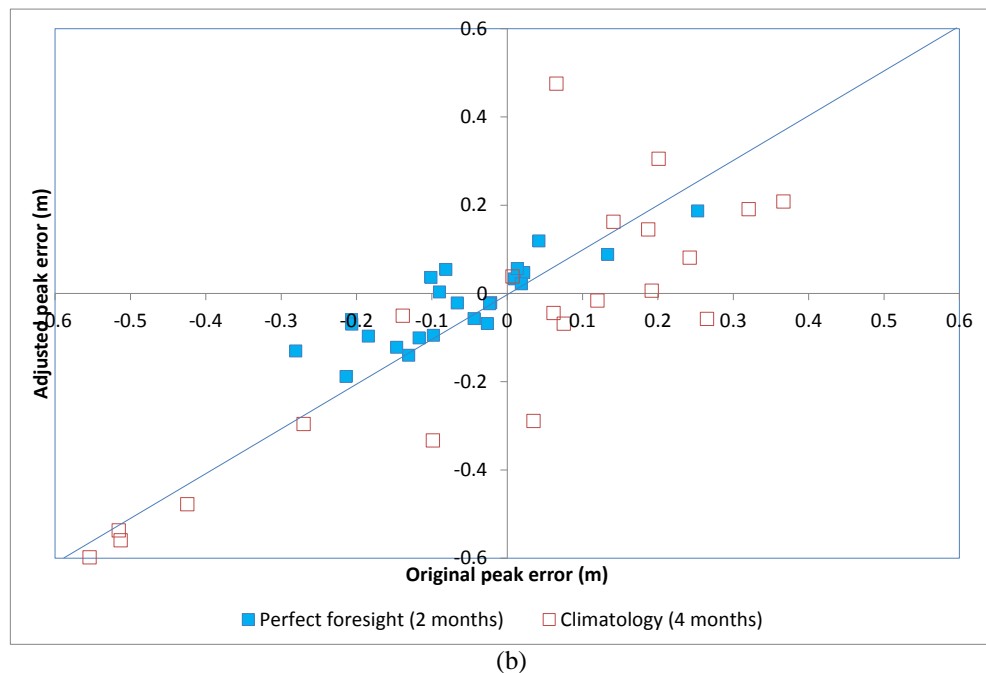

(b)

**Figure 7.** Illustration of the effect of an ARMA approach for the full records using (6,5) models with climatological inputs
and (3,2) models with perfect foresight for (a) Lake Victoria and (b) Lake Malawi with a one-to-one trend line as a guide

For the case of perfect foresight, some improvement was obtained in most years for the Lake Malawi record, with results
more mixed for the climatological estimates, whilst the performance was similar in both cases for the Lake Victoria record,
albeit using a shorter lead time for the climatological estimates. More generally, for these examples, it seemed as if the effect
of these adjustments tails off significantly at lead times of more than about 2-3 months for the Lake Victoria records and 3-4
months for the Lake Malawi records. To improve the results, one option might be to optimise the model orders separately for
each lake record, perhaps using a two stage approach in which the climate index adjustments are made first. Depending on
the application, additional performance measures might also be considered to better understand where to focus effort, such as
taking account of the timing differences between peak values or using threshold-based metrics. However these issues were
not pursued further since, as noted in the following section, for operational forecasting alternative inputs would be used, each
with their own error characteristics and bias correction requirements.

**4 Discussion and conclusions**

The aim here has been to illustrate an approach to exploring possible model structures considering factors such as the choice of input variables, characteristic response times and options for data assimilation. A mixture of transfer function, regression and analytical techniques was used. In keeping with this overall approach, some aspects were only developed to the stage required to draw useful conclusions rather than providing a full solution, such as with the net inflow model and data assimilation components. In future studies, these types of analyses might then help to guide the development of more complex models for individual lakes, or groups of lakes.

For the two lakes considered here, some initial findings include the possibility of using lake rainfall alone as a model input, and the potential to use error prediction techniques that are more typical of those used for short-range flow forecasting, combined with statistical relationships incorporating climate indices. Regarding seasonal variations in levels, due to storage influences there was some evidence of forecast skill up to 3-6 months ahead solely from climatological rainfall estimates, depending on the performance measures used. For the Lake Victoria record, as found in several previous studies, there were also indications of a change in response following the extreme rainfall events of the early 1960s. In any further studies, though, for these or other lakes, the results would need to be evaluated using contemporary datasets taking account of both uncertainties in the observations and, for regulated lakes, current operating rules via an appropriate parameterisation of outflow relationships, including allowing for any changes in procedures over time.

For example, one particularly difficult decision is on the choice of rainfall records to use, and whether these should be area-averaged values (as here) or index series from representative gauges. As already noted, some measurement challenges include the spatial coverage of gauges and any local enhancement of rainfall due to the influences on atmospheric circulation from the lake. Alternatively, where raingauge networks are sparse, satellite observations provide another possibility, although with their own measurement challenges such as the need to differentiate between locally-driven convective and stratiform rainfall and land and water surfaces. With raingauge inputs, another potential challenge is the need for data-sharing agreements when gauges are operated by more than one organisation or country.

More generally, though, estimates for lake rainfall should in principle be more accurate than for catchment rainfall since there are no topographic influences to consider - other than around the lake shoreline - and for that reason were selected here. However, given the huge areas covered, the challenges in estimating the individual components in the water balance should not be underestimated, and for forecasting purposes these are perhaps best regarded as index series themselves. The uncertainty in the estimates then cascades into the water balance estimates and hence level and outflow forecasts. For model calibration, this emphasises the importance of water balance studies in evaluating the suitability of any inputs which are proposed for operational use. Also for smaller lakes with faster response times, a weekly or even daily time step might be required to capture the main features of the response, particularly for tributary inflows. However, this brings new modelling challenges such as the need to consider the spatial variability in different components of the water balance in more detail,

and the timing differences between them. Of course, if links to rainfall are not of interest, there is the option of formulating models directly in terms of the net inflows derived from levels and outflows, which are often easier to estimate.

Regarding rainfall forecasts, these offer the potential to extend lead times further through direct input of ensemble rainfall forecasts, perhaps combined with the seasonal forecasts for climate indices which are now also routinely available. For the

two lakes considered, the focus on model development to date has been for Lake Victoria, including operational tools for hazardous thunderstorms (Thiery et al. 2016, 2017) and research studies on developing high resolution models for seasonal forecasting (Argent et al. 2014). The performance of global scale models for seasonal forecasting has also been evaluated in a regional context suggesting that these offer some improvements on the consensus forecasts prepared by Regional Climate Outlook Forums (Mwangi et al. 2014). Promising results have also been obtained for drought forecasting at seasonal

timescales in southern Africa (Winsemius et al. 2014).

For the development of seasonal models, the reforecasts available from sources such as the Subseasonal to Seasonal (S2S) Prediction Project (Vitart et al. 2017) provide a valuable resource. So-called custom climate indices or predictors might also be considered based on additional meteorological and ocean parameters; for example using relationships derived from principal component analyses or a transfer function approach.

Another consideration is the modelling approach to use and this will typically depend on the operational requirement and the real-time data available, and to some extent the skills and preferences of the modelling team. For example, some additional factors to consider might possibly include artificial influences on lake inflows or outflows from hydropower or other operations, and whether modelling components are required for water quality, ecology, snowmelt and sediment transport.

The classical approach to modelling a lake water balance is to use rainfall-runoff models to estimate tributary inflows, with separate components for area-averaging of lake and catchment rainfall and the outflow response. The runoff components are typically estimated in semi-distributed or distributed form using a conceptual or physical-conceptual approach, and lake evaporation is typically estimated from local weather station records or an energy budget approach.

In contrast, if a transfer function modelling approach is adopted, the step-by-step approach illustrated here provides a

powerful way to rapidly explore many options. However this does not fully exploit the power of the stochastic techniques used, particularly regarding the use of time varying parameters: a concept from system engineering in which forecasts for the parameters themselves become part of the solution, and a way of reflecting potential long-term trends and variations in climate. In that regard, it is worth noting that the water balance itself can be solved in transfer function form with no further approximation for the case of discrete time intervals (as here) and a linear outflow relationship (b=1).

For the more general non-linear case, the same approach can still be adopted but the model coefficients now become a function of the levels or - to use a phrase common in system engineering - are 'state dependent' (e.g. Young 1984). Although this would be considerably more complex than was required in this case, an overall solution could be envisaged in which the net inflow model, water balance and data assimilation components are combined using a so-called State Dependent Parameter (SDP) approach. Some aspects of this approach have already been illustrated by the methods described

here and there is an extensive literature regarding more complete solutions (e.g. Young, 2000, Sadeghi et.al. 2010), including flood forecasting applications (e.g. Young and Beven, 1994; Leedal et al., 2013; Smith et al. 2014).

## Acknowledgements

We would like to thank the Centre for Ecology and Hydrology for permission to use the datasets for Lake Victoria; also Michael Kizza for providing additional independently derived records for comparison. The WMO (1983) study was led by Dr Chris Kidd and provides a valued contribution to understanding of the hydrology of Lake Malawi. The ENSO index datasets were provided by the NOAA/OAR/ESRL PSD, Boulder, Colorado, USA, from their Web site at http://www.esrl.noaa.gov/psd/, and the IOD dataset was provided by Application Laboratory (APL)/JAMSTEC. The present analyses were performed whilst the lead author was an Honorary Researcher at Lancaster Environment Centre. We are grateful to M-H.Ramos, W.Thiery and K.Engeland for their comments on the first draft of this paper.

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
