# Peer review of "Exploratory studies into seasonal flow forecasting potential for large lakes"

_Hydrology and Earth System Sciences, 2017_

## Referee Comment (RC1) · K. Engeland (Referee) · 22 Jun 2017

The title explains well the content of the paper: "Exploratory studies into seasonal flow forecasting potential for large lakes". I think the paper is interesting and provide some useful insights and conclusions.

First a general comment on the evaluation of model performance in this study. When assessing forecasting skill, the benchmark that is used as a reference for assessing skill should be given. If there is a strong seasonality in lake outflows, maybe a monthly climatology would be a better benchmark than a long term average. See e.g. Bettina Schaefli and Hoshin V. Gupta (2007) for choosing benchmark in catchments with a strong seasonality in runoff. Another performance measure that could be used is

anomaly correlation coefficient.

The paper is, in general, well written, but some parts of the manuscript could benefit from more clarity in the presentation. I will give some suggestions below.

Introduction

The introduction is rather brief, and it could be useful to refer to both operational systems and research papers describing approaches that are used for seasonal forecasting of lake levels or outflow. E.g. for the great lakes in US/Canada there is an operational seasonal forecasting service: https://www.glerl.noaa.gov/data/wlevels/levels.html#modelsAndForecasts. In particular if there are other studies on forecasting the water levels in lake Victoria and Malawi could be useful. One recent example is Mulumpwa et al. (2017). I also think the introduction could better reflect the content of the paper, in particular the use of different circulation indices as a predictors for forecasting water levels. Maybe small parts of the data section could be moved to the introduction. In the end of the introduction I miss some clearly stated aims or objectives.

Case Studies

Often it is challenging to estimate outflows based on time series of lake levels since the results might be very sensitive to quality of water level observations. In particular for large lakes where one mm water level represents a large volume, using this approach for daily values, results in a lot of noise. It helps to use weekly or monthly values as in this study.

I miss a more specific description of the data: What is the time resolution of water level and outflow data you used?

Methodology

It would be useful if you in the methods section explains more explicitly the combination of models that yare used, i.e. how is the net inflow model combined with the lake

response model. Further on, how are the regression and ARMA models are used, i.e. is the the residuals of the lake response model the dependent variable?

Results

The previous comment on telling what is used as dependent variable in the regression and the ARMA modelling is important when presenting results on lines 12-16 on page 15. Section 3.2 "Net inflow estimates" is maybe not very precise. As I understand, you want to use this model as a simple forecasting model where forecasted precipitation is used to drive the model. Would "Net inflow forecasting" be a better sub-title?

Discussion

Many of the great lakes are located in areas with a seasonal snov cover. In the introduction seasonal forecasting in snow dominated catchments is mentioned, but it could be useful to speculate on seasonal forecasting for catchments with large lakes and seasonal snow cover.

Figures

Figure 2: could change the scale of the y-axis to be between 0.5 and 1.5

Figure 4: It is difficult to see the observations (the dots). It also seems like the 95% confidence intervals are too wide since all observations are well inside this interval. Please comment.

Figure 5: It is difficult to see the difference between the lines.

Equations

Equation 1 and 2: I have some questions about dimensions in these equations. On the left hand side, dh/dt has the dimension length/time, so then N should also have the same dimension. Then "depth per unit area of lake surface" is confusing. I suggest to use "Volume flux per unit area of lake surface". In Equation 2 I miss a $\Delta t$. Either should (i) the fluxes P, E, Qc and Q0 be integrated over the time interval $\Delta t$ in order to

become water depths, or (ii) Δh be divided by Δt in order to become a flux. If the latter alternative is used, it could be useful to state that all P, E, Qc , Q0 are average fluxes over the time interval Δt.

Equation 7 and 8: it could be useful to avoid using a and b here since these symbols are already used in Equation 3.

Equation 7: What is n and yt?

Equation 9: It is is difficult to understand this equation. What is A and B? previously A was used for lake surface area.

Equation 12: This equation is not necessary.

References:

Bettina Schaefli and Hoshin V. Gupta (2007) Do Nash values have value? Hydrol. Process. 21, 2075–2080 (2007) DOI: 10.1002/hyp.6825

M. Mulumpwa, W. W. L. Jere, M. Lazaro, A. H. N. Mtethiwa n(2017) World Academy of Science, Engineering and Technology, International Journal of Environmental and Ecological Engineering Vol:4, No:3, 2017 Forecasting Lake Malawi Water Level Fluctuations Using Stochastic Models

---

## Author Comment (AC1) · 7 Jul 2017

**Exploratory studies into seasonal flow forecasting potential for large lakes**

**Kevin Sene, Wlodek Tych, Keith Beven**

**Author response to interactive comment by K. Engeland, July 2017**

*The title explains well the content of the paper: "Exploratory studies into seasonal flow forecasting potential for large lakes". I think the paper is interesting and provide some useful insights and conclusions.*

Thanks for these useful comments. We have provided some responses below including items that we will change when submitting the revised version of the paper

*First a general comment on the evaluation of model performance in this study. When assessing forecasting skill, the benchmark that is used as a reference for assessing skill should be given. If there is a strong seasonality in lake outflows, maybe a monthly climatology would be a better benchmark than a long term average. See e.g. Bettina Schaefli and Hoshin V. Gupta (2007) for choosing benchmark in catchments with a strong seasonality in runoff. Another performance measure that could be used is anomaly correlation coefficient.*

We agree and, when looking over long time periods, as we note another issue is the non-stationarity in outflows; for example as illustrated by the step change in observed values for Lake Victoria in the early 1960s. We therefore decided to focus on estimates for the errors in annual peak values (Figures 6 and 7) since peaks tend to occur around the same time each year and are the most challenging to forecast. This would then avoid choice of a specific measure, or measures, although again as noted does not take account of timing errors. For the revised paper, we will include some more discussion of these points and possible alternative measure that could be used (e.g. as described by Schaefli and Gupta 2007).

*The paper is, in general, well written, but some parts of the manuscript could benefit from more clarity in the presentation. I will give some suggestions below.*

**Introduction**

*The introduction is rather brief, and it could be useful to refer to both operational systems and research papers describing approaches that are used for seasonal forecasting of lake levels or outflow. E.g. for the great lakes in US/Canada there is an operational seasonal forecasting service: https://www.glerl.noaa.gov/data/wlevels/levels.html#modelsAndForecasts. In particular if there are other studies on forecasting the water levels in lake Victoria and Malawi could be useful. One recent example is Mulumpwa et al. (2017). I also think the introduction could better reflect the content of the paper, in particular the use of different circulation indices as a predictors for forecasting water levels. Maybe small parts of the data section could be moved to the introduction. In the end of the introduction I miss some clearly stated aims or objectives.*

This is a valid point and we will extend the introduction to include more discussion of seasonal forecasting approaches for other lakes and, as suggested, move some of the data section forwards. Thanks also for the Lake Malawi citation which we had not yet seen

**Case Studies**

*Often it is challenging to estimate outflows based on time series of lake levels since the results might be very sensitive to quality of water level observations. In particular for large lakes where one mm water level represents a large volume, using this approach for daily values, results in a lot of noise. It helps to use weekly or monthly values as in this study. I miss a more specific description of the data: What is the time resolution of water level and outflow data you used?*

Sorry if this was not clear; the time resolution used was monthly. We will clarify this in the revised version

**Methodology**

*It would be useful if you in the methods section explains more explicitly the combination of models that are used, i.e. how is the net inflow model combined with the lake response model. Further on, how are the regression and ARMA models are used, i.e. is the residuals of the lake response model the dependent variable?*

Again we will clarify this in the revised paper (and yes the residuals were the dependent variable)

**Results**

*The previous comment on telling what is used as dependent variable in the regression and the ARMA modelling is important when presenting results on lines 12-16 on page 15.*

Agreed, we will include further discussion of this point

*Section 3.2 "Net inflow estimates" is maybe not very precise. As I understand, you want to use this model as a simple forecasting model where forecasted precipitation is used to drive the model. Would "Net inflow forecasting" be a better sub-title?*

Agreed, we will change this in the revised version

**Discussion**

*Many of the great lakes are located in areas with a seasonal snov cover. In the introduction seasonal forecasting in snow dominated catchments is mentioned, but it could be useful to speculate on seasonal forecasting for catchments with large lakes and seasonal snow cover.*

This is not something that we looked at but we are happy to include some speculation regarding this point

**Figures**

*Figure 2: could change the scale of the y-axis to be between 0.5 and 1.5*

Agreed, we will change this

*Figure 4: It is difficult to see the observations (the dots). It also seems like the 95% confidence intervals are too wide since all observations are well inside this interval.*

This was just a small part of the record so is perhaps not representative of the overall performance; we will revisit this and include a different figure and/or a comment in the revised version

Please comment.

*Figure 5: It is difficult to see the difference between the lines.*

The figure will be improved

**Equations**

*Equation 1 and 2: I have some questions about dimensions in these equations. On the left hand side, dh/dt has the dimension length/time, so then N should also have the same dimension. Then "depth per unit area of lake surface" is confusing. I suggest to use "Volume flux per unit area of lake surface". In Equation 2 I miss a t. Either should (i) the fluxes P, E, Qc and Q0 be integrated over the time interval t in order to become water depths, or (ii) h be divided by t in order to become a flux. If the latteralternative is used, it could be useful to state that all P, E, Qc , Q0 are average fluxes over the time interval t.*

We agree that the description could be improved and will do that in the revised version

*Equation 7 and 8: it could be useful to avoid using a and b here since these symbols are already used in Equation 3.*

Agreed, this will be changed

*Equation 7: What is n and yt?*

Again, this will be clarified

*Equation 9: It is is difficult to understand this equation. What is A and B? previously A was used for lake surface area.*

We agree that this could be confusing; however by convention A is used both for lake areas and in this general form of the transfer function equation. However in the latter case A and B are sometimes written in a bold type face so that is what we propose here

*Equation 12: This equation is not necessary.*

Yes, this will be removed

**References:**

*Bettina Schaefli and Hoshin V. Gupta (2007) Do Nash values have value? Hydrol. Process. 21, 2075–2080 (2007) DOI: 10.1002/hyp.6825*

*M. Mulumpwa, W. W. L. Jere, M. Lazaro, A. H. N. Mtethiwa n(2017) World Academy of Science, Engineering and Technology, International Journal of Environmental and Ecological Engineering Vol:4, No:3, 2017 Forecasting Lake Malawi Water Level Fluctuations using Stochastic Models*

Thanks for these two very useful references; we will include citations and a brief discussion of the findings they describe

---

## Referee Comment (RC2) · W. Thiery (Referee) · 9 Jul 2017

**Review of "Exploratory studies into seasonal flow forecasting potential for large lakes", by Sene et al.**

In this study, the authors explore novel ways to forecast lake levels and outflows that move beyond the classic approach of estimating all individual terms of the lake water balance. The authors discover the potential of combining system inertia (i.e. serial correlation), over-lake precipitation and climate indices for producing skilful seasonal flow predictions.

It seems that this paper is indeed among the first to systematically explore the potential of seasonal flow forecasting in the African Great Lakes regions, a much needed work. As far as I know, also the development of the theoretical framework is innovative and the results of this analysis are interesting. This paper thus demonstrates the potential to make a useful contribution to the scientific literature on seasonal forecasting and the African great Lakes. However, I have some concerns which require the manuscript to be revised. In general the study can only be considered for publication if the comments specified here below are sufficiently addressed.

**General Comments**

1.  My main concern regards the quality of the observational data that are used in this study. As I've worked with meteorological observations in the region extensively myself, I am well aware that these data are often very sparse, and when they are available their quality is often questionable. In general, it would be useful if the authors could discuss whether the quality of the input data may influence their results. The discussion section on page 19 already hints in this direction. See also specific comments 3 – 5 and 8.

**Specific comments**

1.  P3L16: This statement would benefit from a reference

2.  P6L6: In the case of Lake Victoria these departures from the 'Agreed Curve' have been substantial in the past, and a reason for concern. See for Instance Kull (2006), who attributed more than half of the severe drop in lake level during 2004-2005 to over-releases at the Jinja dam.

3.  P7L13: Which data for over-lake precipitation was employed? Precipitation on Lake Victoria is to my knowledge not observed directly except for some sparse data reported by Datta, 1981. Yet we do know that precipitation is highly variable over the lake both in space and time due to the interplay of mesoscale and synoptic-scale circulation (see for instance presentations.copernicus.org/EGU2017-18256_presentation.pptx).

Generally the presence of the African Great Lakes doubles precipitation amounts over their surface area and triggers severe thunderstorms, and shoreline stations are therefore believed to be an inaccurate source of over-lake precipitation (Thiery et al., 2015; 2016). Yet most water balance studies for Lake Victoria employ shoreline stations to estimate over-lake rainfall amounts (Also Piper et al., 1986 who use 8 shoreline stations).

4. P7L15: how many offshore stations were used to estimate precipitation over Lake Malawi? As this lakes stretches >500 km in N-S direction, precipitation patterns may highly differ in space.

5. P7L18: To my knowledge some tributaries are not monitored at all (cfr. 'ungauged perimeter' in Piper et al., 1986).

6. Figure 1: including a simple map showing the locations of Lake Malawi and Lake Victoria as inset in this figure would be useful for the readers not familiar with the study area.

7. Figure 2: As this figure will probably be reduced in size in the final lay-out, it is perhaps useful to shorten the y-axis range.

8. P10L6: are these observations independent from each other? cfr. specific comment 4.

9. P12L10: Did the authors run the forecast model starting every month of the validation period and then computed the $R^2$ per lead time month for the whole period? This is not clear from the method section.

10. P13L13: What is the mechanism behind the strong increase in lake levels leading to higher forecast skill?

11. P13L20: A philosophical point: for short lead times I agree, but for longer lead times it may be impossible to increase the skill of (seasonal) forecasts beyond a certain level. At the moment the meteorological forecast community advances short-term forecast skill at a rate of 1 day per decade of research and technological development…

12. Figure 5: Please replace the y-axis label by '$R^2$' and increase the label font sizes. It may also be useful to add the lake name to each figure panel.

13.    Figure 6: I would mention the climate indices that were used in the regression in the caption of this figure.

14.    P19L17-18: I fully agree; for our new statistical storm warning system for Lake Victoria this approach proves to be highly valuable.

**Textual comments**

1.    P2L1: Please consider changing to "Another situation where storage influences are…"

2.    P3L22: Please consider changing to "if – as is often the case – the"

3.    P3L23: Please consider changing to "rainfall, then"

4.    P3L25: Please consider changing to "the current exploratory studies"

5.    P4L17: Please introduce acronym and perhaps also a reference.

6.    P9L10: Please consider omitting "as noted earlier" to enhance readability.

7.    P12L4: Please consider changing to "records, respectively"

8.    P15L26: please enhance the readability of this sentence

**References**

- Datta, R. K., 1981: Certain aspects of monsoonal precipitation dynamics over Lake Victoria. Monsoon Dynamics, J. Lighthill and R. P. Pearce, Eds., Cambridge University Press, 333–349.
- Kull, D., 2006: Connections Between Recent Water Level Drops in Lake Victoria, Dam Operations and Drought, https://www.internationalrivers.org/sites/default/files/attached-files/full_report_pdf.pdf.
- Piper, B. S., Brown, E., & Sutcliffe, J. V., 1986: The water balance of Lake Victoria. Hydrological Sciences Journal, 58, 342–353.
- Thiery, W., Davin, E.L., Panitz, H.-J., Demuzere, M., Lhermitte, S., van Lipzig, N.P.M., 2015: The impact of the African Great Lakes on the regional climate, J. Climate, 28(10), 4061-4085.
- Thiery, W., Davin, E.L., Seneviratne, S.I., Bedka, K., Lhermitte, S., van Lipzig, N.P.M., 2016: Hazardous thunderstorm intensification over Lake Victoria, Nature Communications, 7, 12786.

---

## Author Response (AR1)

[revised manuscript text omitted]

**Changes made to paper in response to interactive comment by K. Engeland, July 2017**

*The title explains well the content of the paper: "Exploratory studies into seasonal flow forecasting potential for large lakes". I think the paper is interesting and provide some useful insights and conclusions.*

Thanks again for these useful comments; the following notes describe the changes made to the draft paper in response. Please also refer to the initial author comments submitted on 7 July 2017. We have also taken the opportunity to make some minor improvements to wording in the revised version.

*First a general comment on the evaluation of model performance in this study. When assessing forecasting skill, the benchmark that is used as a reference for assessing skill should be given. If there is a strong seasonality in lake outflows, maybe a monthly climatology would be a better benchmark than a long term average. See e.g. Bettina Schaefli and Hoshin V. Gupta (2007) for choosing benchmark in catchments with a strong seasonality in runoff. Another performance measure that could be used is anomaly correlation coefficient.*

To help address this comment the text has been changed as follows (see the response to comments from Reviewer 2 also):

A new paragraph has been added after Line 16 on Page 15 as follows:

"However, the use of $R^2$ only provides one view of performance and, as already noted, has some limitations. Also, lake levels are strongly seasonal and there might therefore be a case to use alternative benchmarks such as seasonally-based values or metrics which incorporate lag times or moving averages as proposed by Schaefli and Gupta (2007). Indeed there are many possible measures which could be used (e.g. Jolliffe and Stephenson 2011, Wilks 2011) and this is a topic that would merit further research. For these exploratory studies, though, a simpler approach was adopted which was to focus on the errors in annual peak levels since these tend to occur around the same time each year and are more challenging to forecast than annual minima."

The paragraph beginning on Line 22 of Page 13 now begins:

"These results of course only give one view of performance and, given the findings from previous climate studies for these lakes, and the cross correlation estimates from the present study, one possibility is that the errors in model outputs might to some extent…"

*The paper is, in general, well written, but some parts of the manuscript could benefit from more clarity in the presentation. I will give some suggestions below.*

**Introduction**

*The introduction is rather brief, and it could be useful to refer to both operational systems and research papers describing approaches that are used for seasonal forecasting of lake levels or outflow. E.g. for the great lakes in*

*US/Canada there is an operational seasonal forecasting service: https://www.glerl.noaa.gov/data/wlevels/levels.html#modelsAndForecasts. In particular if there are other studies on forecasting the water levels in lake Victoria and Malawi could be useful. One recent example is Mulumpwa et al. (2017). I also think the introduction could better reflect the content of the paper, in particular the use of different circulation indices as a predictors for forecasting water levels. Maybe small parts of the data section could be moved to the introduction. In the end of the introduction I miss some clearly stated aims or objectives.*

We have changed the emphasis of the introduction slightly as suggested, including giving more references to previous studies and moving some of the discussion on climate indices forwards from the main text. The last three paragraphs of the introduction now read:

"For large lakes, perhaps the longest-established operational systems for seasonal forecasting of lake levels are for the Great Lakes in the USA and Canada. Both empirical techniques and physically-based hydrological models are used driven by long-range climate outlooks for precipitation and air temperatures with, more recently, incorporation of ensemble outputs from regional climate models (Gronewold et al. 2011, Bollinger et al. 2017). On a smaller scale, of course, water balance models for reservoirs and regulated lakes often form part of regional water supply models typically using statistical techniques or ensemble streamflow prediction approaches (Yeh 1985, Pagano et al. 2009). The potential for seasonal forecasting using statistical models has also been explored for the rivers downstream of two of the largest lakes in Africa: Lake Malawi (Jury 2014) and Lake Victoria (Siam and Eltahir 2015) and – over longer time scales - Mulumpwa et al. (2017) derived probabilistic estimates for future levels of Lake Malawi using a univariate stochastic model.

Here we describe exploratory studies into forecasting potential for large lakes using an alternative approach. Following a brief review of the dynamic characteristics of lake response, both stochastic dynamic regression and transfer function approaches are used to explore the relationships between a range of potential predictors and lake levels and outflows. This general approach has been widely applied to real-time flood forecasting applications (e.g. Lees 2000, Smith et al. 2013) and, in addition to the ease with which options can be explored, a key advantage is that few prior assumptions are required about the nature of those relationships (e.g. Beven 2009, Young, 2013).

The methods are evaluated using case studies for Lake Victoria and Lake Malawi including the potential benefits of data assimilation in reducing the impacts of measurement and modelling uncertainties. As in the studies by Jury (2014) and Siam and Eltahir (2015) the use of climate indices is considered since both the El Niño Southern Oscillation and Indian Ocean Dipole are thought to have a significant influence on rainfall in east and southern Africa (e.g. Saji et al. 1999, Nicholson and Selato 2000, Jury and Gwazantini 2002, and Manatsa et al. 2011). Since the main aim was to provide insights into possible model structures, the analyses were based primarily on historical datasets derived as part of previous water balance studies as this allowed a more detailed investigation of lake response than would be possible using contemporary datasets. By chance the periods covered included some of the most significant flood and drought periods on record allowing model performance to be evaluated under these more extreme conditions. The discussion concludes with some suggestions for how the findings could be translated into operational forecasting models."

**Case Studies**

*Often it is challenging to estimate outflows based on time series of lake levels since the results might be very sensitive to quality of water level observations. In particular for large lakes where one mm water level represents a large volume, using this approach for daily values, results in a lot of noise. It helps to use weekly or monthly values as in this study. I miss a more specific description of the data: What is the time resolution of water level and outflow data you used?*

The time interval is now explicitly stated in the bullet points on Page 7

**Methodology**

*It would be useful if you in the methods section explain more explicitly the combination of models that are used, i.e. how is the net inflow model combined with the lake response model. Further on, how are the regression and ARMA models are used, i.e. is the residuals of the lake response model the dependent variable?*

The first paragraph of Section 3.3 has been modified to read as follows which is hopefully clearer (see the response to Reviewer 2 also):

"Having developed models for the net inflows, these were expressed in recursive form for input to the water balance equation (Eq. (4). This formulation mimics how the models would be used in an operational setting in which the water balance would be solved numerically each month to derive forecasts for lake levels and outflows for the months ahead based on rainfall observations available up to the time of the forecast. To further extend forecast lead times, rainfall forecasts would ideally also be required but, for these exploratory studies, it was sufficient to use climatological estimates instead, based on the mean monthly distributions of rainfall in the calibration period.  However, to provide an indication of forecast potential, the performance was also estimated assuming perfect foresight of rainfall: that is, using historical observed values beyond the forecast origin. In both cases, the first two years of record were ignored to allow for initialisation of the autocorrelation aspects of the models. "

Line 1 of page 15 has also been modified to read:

"To test this hypothesis, regression models were developed between the residuals of the forecast outputs and the climate indices described earlier…"

and the caption for Figure 6 amended to:

"Figure 6. Illustration of the effect of including fixed parameter regression relationships with DMI and NINO34 for the model residuals on errors in annual maximum levels at lead times of (a) 5 months for Lake Victoria and (b) 4 months for Lake Malawi.  Climatological rainfall inputs were used and a one-to-one trend line is shown as a guide."

**Results**

*The previous comment on telling what is used as dependent variable in the regression and the ARMA modelling is important when presenting results on lines 12-16 on page 15.*

Please see the edits above under 'Methodology'. Line 4 of page 17 has also been modified to read:

However, exploratory studies using an ARMA approach for the residuals…..

*Section 3.2 "Net inflow estimates" is maybe not very precise. As I understand, you want to use this model as a simple forecasting model where forecasted precipitation is used to drive the model. Would "Net inflow forecasting" be a better sub-title?*

The subtitle has been changed to 'Net inflow forecasting'

**Discussion**

*Many of the great lakes are located in areas with a seasonal snow cover. In the introduction seasonal forecasting in snow dominated catchments is mentioned, but it could be useful to speculate on seasonal forecasting for catchments with large lakes and seasonal snow cover.*

We had originally planned to include a short discussion of snow dominated catchments and drafted some text for the discussion section. However on reflection this seemed out of place in a paper about tropical lakes. In case this is useful, though, the text that we drafted was as follows:

"In colder regions where snowfall is a factor, air temperature forecasts could also be used to help indicate the likely timing of snowmelt, and possibly incorporated in an ensemble streamflow prediction approach. An indicator of snowpack depth, such as the snow water equivalent, might be used as another input variable. In addition to snowfall on the land surface, another factor to consider is ice on the lake surface and any associated accumulation of snow over the lake, so additional indicators might be the percentage of the surface covered with ice, plus the typical ice thickness."

We have however mentioned snowmelt in the discussion section as follows:

"….and whether modelling components are required for water quality, ecology, snowmelt and sediment transport."

**Figures**

*Figure 2: could change the scale of the y-axis to be between 0.5 and 1.5*

The y-axis range has been changed as suggested

*Figure 4: It is difficult to see the observations (the dots). It also seems like the 95% confidence intervals are too wide since all observations are well inside this interval.*

The figure has been changed for an example from the validation period with the dots representing observations made larger and some minor changes to the associated text made relating to this

*Figure 5: It is difficult to see the difference between the lines.*

The figure has been replaced with hopefully a clearer version

**Equations**

*Equation 1 and 2: I have some questions about dimensions in these equations. On the left hand side, dh/dt has the dimension length/time, so then N should also have the same dimension. Then "depth per unit area of lake surface" is confusing. I suggest to use "Volume flux per unit area of lake surface". In Equation 2 I miss a t. Either should (i) the fluxes P, E, Qc and Q0 be integrated over the time interval t in order to become water depths, or (ii) h be divided by t in order to become a flux. If the latteralternative is used, it could be useful to state that all P, E, Qc , Q0 are average fluxes over the time interval t.*

The description beneath Equation 1 has been amended to include the phrases:

"The term N is the net inflow, expressed as a volumetric flux per unit area of lake surface…"

and

"…and $Q_c$ is the inflow from the surrounding catchment area, again expressed in terms of unit lake surface area."

*Equation 7 and 8: it could be useful to avoid using a and b here since these symbols are already used in Equation 3.*

The coefficients in Equations 7 and 8 have been changed to r and s to differentiate them from Equation 3 and the associated text modified

*Equation 7: What is n and yt?*

The sentence beneath Equation (7) has been changed to:

 "where $y_t$ is the dependent variable, $r_i$ are the model coefficients, which can be time varying if required, n is the maximum lag time considered…."

*Equation 9: It is is difficult to understand this equation. What is A and B? previously A was used for lake surface area.*

A, B, C and D in Equation 9 are now written in bold face type in the equation and the text that immediately follows the equation now reads:

"where $z^{-1}$ is the backward shift operator ($z^{-1} y_t = y_{t-i}$). The second term represents the residuals of the transfer function input-output model via polynomials **C** and **D** and, although not used for the net inflow component, an Auto Regressive-Moving Average (ARMA) model of this form (e.g. Box and Jenkins 1970) was used in the data assimilation component described later. Here a single external input is considered but as discussed later the formulation is easily extended to multiple inputs."

*Equation 12: This equation is not necessary.*

The equation has been removed and the text updated to:

"In that regard it is worth noting that the water balance itself can be solved in transfer function form with no further approximation for the case of discrete time intervals (as here) and a linear outflow relationship (b=1). For the more general non-linear case, the same approach can still be adopted but the model coefficients now become a function of the levels or - to use a phrase common in system engineering - are 'state dependent' (e.g. Young 1984)."

We have also taken the opportunity to improve a sentence in the previous paragraph as follows:

".....a powerful concept from system engineering in which forecasts for the parameters themselves become part of the solution as a way of reflecting potential long-term trends and variations in climate."

and to include two more references - in addition to Smith et al. 2014 - giving examples of flood forecasting applications in the discussion on SDP approaches:

"Some aspects of this approach have already been illustrated by the methods described here and there is an extensive literature regarding more complete solutions (e.g. Young, 2000, Sadeghi et.al. 2010) including flood forecasting applications (e.g. Young and Beven, 1994; Leedal et al., 2013; Smith et al. 2014).

*In this study, the authors explore novel ways to forecast lake levels and outflows that move beyond the classic approach of estimating all individual terms of the lake water balance. The authors discover the potential of combining system inertia (i.e. serial correlation), over-lake precipitation and climate indices for producing skilful seasonal flow predictions.*

*It seems that this paper is indeed among the first to systematically explore the potential of seasonal flow forecasting in the African Great Lakes regions, a much needed work. As far as I know, also the development of the theoretical framework is innovative and the results of this analysis are interesting. This paper thus demonstrates the potential to make a useful contribution to the scientific literature on seasonal forecasting and the African great Lakes. However, I have some concerns which require the manuscript to be revised. In general the study can only be considered for publication if the comments specified here below are sufficiently addressed.*

Thanks for these useful comments; the following notes describe the main changes made to the draft paper in response. We have also taken the opportunity to make some minor changes to wording in the revised version.

**General Comments**

*1. My main concern regards the quality of the observational data that are used in this study. As I've worked with meteorological observations in the region extensively myself, I am well aware that these data are often very sparse, and when they are available their quality is often questionable. In general, it would be useful if the authors could discuss whether the quality of the input data may influence their results. The discussion section on page 19 already hints in this direction. See also specific comments 3 – 5 and 8.*

We agree with these comments; please refer to the edits noted below

*Specific comments*

*1. P3L16: This statement would benefit from a reference*

We have changed the statement to read as follows and added the citation mentioned to the list of references:

"However, for ungauged catchments it is often found that, on an annual basis, and sometimes a monthly basis, empirical regression relationships can be derived between flow-related parameters of interest and catchment

characteristics such as the area and slope (e.g. Smakhtin 2001). Perhaps the simplest of all such approaches is to assume a mean runoff coefficient r=Qc/(AcPc) in which case the net inflow is given by:"

*2. P6L6: In the case of Lake Victoria these departures from the 'Agreed Curve' have been substantial in the past, and a reason for concern. See for Instance Kull (2006), who attributed more than half of the severe drop in lake level during 2004-2005 to over-releases at the Jinja dam.*

Thanks for mentioning this. We were aware that the history of derivation and application of the Agreed Curve is complex but, since these were exploratory studies, rather than delve into these issues, we therefore took the simple expedient of excluding periods with significant departures from the evaluation. However, to emphasise the challenges further, we have changed the sentence beginning on Line 8 of Page 6 to read:

"For the Lake Victoria studies the periods omitted only amounted to a small part of the record (which predates more recent departures)…."

and the final sentence of the second paragraph in the discussion section to read:

"In any further studies, though, for these or other lakes, the results would need to be evaluated using contemporary datasets taking account of both uncertainties in the observations and, for regulated lakes, current operating rules via an appropriate parameterisation of outflow relationships, including allowing for any changes in procedures over time."

*3. P7L13: Which data for over-lake precipitation was employed? Precipitation on Lake Victoria is to my knowledge not observed directly except for some sparse data reported by Datta, 1981. Yet we do know that precipitation is highly variable over the lake both in space and time due to the interplay of mesoscale and synoptic-scale circulation (see for instance presentations.copernicus.org/EGU2017-18256_presentation.pptx).*

Yes, we agree, particularly for daily and sub-daily rainfall estimates (rather than the monthly values used here). Please see the response to the next comment as well.  We have added a reference to Thiery et al. 2015 in the discussions of lake influences on local climate and changed the sentence beginning on Line 28 of Page 17 to read:

"Also for smaller lakes with faster response times, a weekly or even daily time step might be required to capture the main features of the response, particularly for tributary inflows, although this brings new modelling

challenges such as the need to consider the spatial variability in different components of the water balance in more detail, and the timing differences between them."

*Generally the presence of the African Great Lakes doubles precipitation amounts over their surface area and triggers severe thunderstorms, and shoreline stations are therefore believed to be an inaccurate source of over-lake precipitation (Thiery et al., 2015; 2016). Yet most water balance studies for Lake Victoria employ shoreline stations to estimate over-lake rainfall amounts (Also Piper et al., 1986 who use 8 shoreline stations).*

To provide more background on the challenges and potential ways of mitigating these issues the following text has been included at the end of Section 2.2:

"For the Lake Malawi records, it is worth noting that sometimes a small correction term is included to account for inflows and losses between the lake outlet and Kamuzu Barrage; however the impacts are small when considered on a monthly basis and as here this term is often omitted. As indicated above, values are for a hydrological year of November to October but, to help comparisons with the Lake Victoria analyses, when describing results the predominant calendar year is cited; for example '1970' refers to the hydrological year 1969/70. For the Lake Victoria datasets that were used another point to note is that to help to account for the increased rainfall over the lake surface, the lake rainfall estimates based on raingauge observations were increased using linear scaling factors to preserve an overall water balance between the start and end points of the simulation period, although with no constraints on the variability in the intervening years.

As the many citations above indicate, much has been written about the accuracy of the various gauge records available and the derived water balance components, together with issues such as how rainfall measured at the shoreline relates to average rainfall at the lake surface. For these exploratory studies, to facilitate comparisons, unless otherwise stated values for levels and other parameters were generally expressed in standardised form, based on the departure from the mean divided by the standard deviation in each time period of interest. This also helped to reduce the uncertainties in the results by focussing on the underlying signals rather than being concerned about absolute amounts, such as estimates for the lake rainfall. In forecasting applications, another consideration is that real-time observations can be used to adjust forecasts to help to account for modelling and observation errors and the potential value of this process, called data assimilation or adaptive modelling, is discussed later."

Partly also in response to comments from Reviewer 1, the following sentence has also been included in the introduction regarding the potential role of data assimilation:

"The methods are evaluated using case studies for Lake Victoria and Lake Malawi including the potential benefits of data assimilation in reducing the impacts of measurement and modelling uncertainties."

Also, the bullet point relating to Lake Victoria records on Page 7 has been changed to:

"•      Lake Victoria – monthly estimates from 1925-1978 reported by Piper et al. (1986) and subsequently updated by Institute of Hydrology (1994) to the period 1925-1990 for lake rainfall (6-8 shoreline raingauges), catchment rainfall (25-30 raingauges) and tributary inflows (up to 20 river gauges) and for 1925-1992 for levels and outflows"

and the following text added immediately the two bullet points:

"The numbers in brackets indicate the approximate numbers of gauges used to derive each component in the water balance in these studies with ranges provided in some cases due to differing numbers of gauges in different periods; in the Lake Victoria studies, rainfall-runoff models were also used to estimate inflows for ungauged catchments and to help infill missing values whilst, for Lake Malawi, scaling and correlation approaches were used."

The paragraph on rainfall estimates in the discussion section has also been changed to:

"For example, one particularly difficult decision is on the choice of rainfall records to use, and whether these should be area-averaged values (as here) or index series from representative gauges. As already noted, some measurement challenges include the spatial coverage of gauges and any local enhancement of rainfall due to the influences on atmospheric circulation from the lake. Alternatively, where raingauge networks are sparse, satellite observations provide another possibility, although with their own measurement challenges such as the need to differentiate between locally-driven convective and stratiform rainfall and land and water surfaces. With raingauge inputs, another potential challenge is the need for data-sharing agreements when gauges are operated by more than one organisation or country."

*4. P7L15: how many offshore stations were used to estimate precipitation over Lake Malawi? As this lakes stretches >500 km in N-S direction, precipitation patterns may highly differ in space.*

The relevant bullet point on Page 7 has been modified to:

"•     Lake Malawi – monthly estimates for the period November 1954 to October 1980 reported by WMO (1983) for lake rainfall (16 shoreline raingauges; 1 island gauge), catchment rainfall (53 raingauges) and tributary inflows (about 21 river gauges), and for which aspects of the water balance appear in a number of the studies cited here, such as Drayton (1984) and Neuland (1984)"

*5. P7L18: To my knowledge some tributaries are not monitored at all (cfr. 'ungauged perimeter' in Piper et al., 1986).*

Please see the previous two comments

*6. Figure 1: including a simple map showing the locations of Lake Malawi and Lake Victoria as inset in this figure would be useful for the readers not familiar with the study area.*

Figure 1 now includes a map of the locations of Lake Victoria and Lake Malawi as an inset

*7. Figure 2: As this figure will probably be reduced in size in the final lay-out, it is perhaps useful to shorten the y-axis range.*

The figure has been revised using the axis ranges suggested by Reviewer 1

*8. P10L6: are these observations independent from each other? cfr. specific comment 4.*

Please refer to the comments earlier relating to the numbers of gauges used

*9. P12L10: Did the authors run the forecast model starting every month of the validation period and then computed the R² per lead time month for the whole period? This is not clear from the method section.*

The start of the first paragraph in Section 3.3 has been changed to as follows to hopefully make the methodology clearer (see the response to comments from Reviewer 1 also):

"Having developed models for the net inflows, these were expressed in recursive form for input to the water balance equation (Eq. (4). This formulation mimics how the models would be used in an operational setting in which the water balance would be solved numerically each month to derive forecasts for lake levels and

outflows for the months ahead based on rainfall observations available up to the time of the forecast. To further extend forecast lead times, rainfall forecasts would ideally also be required but, for these exploratory studies…."

Also, the second paragraph in that section now begins:

Figure 5 shows the estimated variations in $R^2$ with lead time for the validation periods for both lakes, derived by advancing the forecast origin by one month between each model run and then retrospectively estimating overall values at the required lead times.

*10. P13L13: What is the mechanism behind the strong increase in lake levels leading to higher forecast skill?*

To hopefully explain this, the following sentence has been added after the sentence that begins on Line 13 on Page 13:

"This is simply an artefact of this type of performance measure since using mean values as a reference becomes less informative if a time series is non-stationary, such as exhibiting trends over time and/or quasi-step changes in values as here."

*11. P13L20: A philosophical point: for short lead times I agree, but for longer lead times it may be impossible to increase the skill of (seasonal) forecasts beyond a certain level. At the moment the meteorological forecast community advances short-term forecast skill at a rate of 1 day per decade of research and technological development…*

In the discussion section we have now included the following additional text with the associated references at the end of the paper:

"For the two lakes considered, the focus on regional model development to date has been for Lake Victoria, including operational tools for hazardous thunderstorms (Thiery et al. 2016, 2017) and research studies on developing high resolution models for seasonal forecasting (Argent et al. 2014).  The performance of global scale models for seasonal forecasting has also been evaluated in a regional context suggesting that these offer some improvements on the consensus forecasts prepared by Regional Climate Outlook Forums (Mwangi et al. 2014). Promising results have also been obtained for drought forecasting at seasonal timescales in southern Africa (Winsemius et al. 2014)."

*12. Figure 5: Please replace the y-axis label by 'R²' and increase the label font sizes. It may also be useful to add the lake name to each figure panel.*

The suggested changes have been made to Figure 5

*13. Figure 6: I would mention the climate indices that were used in the regression in the caption of this figure.*

The caption now begins:

"Illustration of the effect of including fixed parameter regression relationships with DMI and NINO34 for the model residuals on errors in annual maximum levels…."

*14. P19L17-18: I fully agree; for our new statistical storm warning system for Lake Victoria this approach proves to be highly valuable.*

Thanks. As noted earlier we have include citations to Thiery et al. 2016, 2017 in the revised text

*Textual comments*

*1. P2L1: Please consider changing to "Another situation where storage influences are…"*

Text changed as suggested

*2. P3L22: Please consider changing to "if – as is often the case – the"*

Text changed as suggested

*3. P3L23: Please consider changing to "rainfall, then"*

Comma added as suggested

*4. P3L25: Please consider changing to "the current exploratory studies"*

Text changed as suggested

*5. P4L17: Please introduce acronym and perhaps also a reference.*

The text has been amended to:

"….an Auto Regressive-Moving Average (ARMA) model of this form (e.g. Box and Jenkins 1970)…"

And the Box and Jenkins reference added to the list of references

*6. P9L10: Please consider omitting "as noted earlier" to enhance readability.*

Phrase omitted as suggested

*7. P12L4: Please consider changing to "records, respectively"*

Comma added as suggested

*8. P15L26: please enhance the readability of this sentence*

The sentence is now as follows:

"However for both records further investigation would be required into the optimum approach to use; for example exploring the influence of timing errors on model performance and whether time varying parameters could be used to help represent longer term trends and other variations."

---

## Author Response (AR2)

**Exploratory studies into seasonal flow forecasting potential for large lakes**
**Kevin Sene, Wlodek Tych, Keith Beven**
**Editor Decision – technical corrections – author comments**

Thanks for the comments; as a result the following changes have been made to the manuscript:

- Fig. 4 – caption amended to note sample uncertainties
- Punctuation (commas) – minor changes made throughout the document
- Page 14, lines 3-7 – wording hopefully improved
- Page 14, line 14 – wording changed as suggested
- Page 16, line 16 – sentence split as suggested
- Page 19, line 19 – period added to end of sentence